# A Modular Multi-task Reasoning Framework Integrating Spatio-temporal Models and LLMs

## Abstract

Spatio-temporal data mining plays a pivotal role in informed decision making across diverse domains. However, existing models are often restricted to narrow tasks, lacking the capacity for multi-task inference and complex long-form reasoning that require generation of in-depth, explanatory outputs. These limitations restrict their applicability to real-world, multi-faceted decision scenarios. In this work, we introduce STReason, a novel framework that integrates the reasoning strengths of large language models (LLMs) with the analytical capabilities of spatio-temporal models for multi-task inference and execution. Without requiring task-specific finetuning, STReason leverages in-context learning to decompose complex natural language queries into modular, interpretable programs, which are then systematically executed to generate both solutions and detailed rationales. To facilitate rigorous evaluation, we construct a new benchmark dataset and propose a unified evaluation framework with metrics specifically designed for long-form spatio-temporal reasoning. Experimental results show that STReason significantly outperforms advanced LLM baselines across all metrics, particularly excelling in complex, reasoning-intensive spatio-temporal scenarios. Human evaluations further validate STReason's credibility and practical utility, demonstrating its potential to reduce expert workload and broaden the applicability to real-world spatio-temporal tasks. We believe STReason provides a promising direction for developing more capable and generalizable spatio-temporal reasoning systems. Our code is available at: `https://anonymous.4open.science/r/STReason-B0B2/`

## 1 Introduction

In the realm of data science, spatio-temporal data, characterized by both spatial and temporal dimensions, plays a critical role in a wide array of fields such as environmental monitoring (Hettige et al., 2024; Liang et al., 2023), urban planning and traffic management (Li et al., 2024b; Ji et al., 2022), and public health (Dong et al., 2024). Over the years, research in spatio-temporal data mining has progressed from conventional statistical and machine learning approaches (Xie et al., 2020) to advanced deep learning frameworks (Jin et al., 2023a; Wang et al., 2020; Zhang et al., 2024). In recent years, the development of Foundation Models (FMs) has sparked a surge in research aimed at improving spatio-temporal modeling through Large Language Models (LLMs) (Li et al., 2024b; Liang et al., 2024; 2025). By leveraging the strengths of LLMs in generalization, cross-modal reasoning, and long-sequence modeling, several LLM-based spatio-temporal models have been developed for various applications with notable performance improvements in zero-shot and few-shot scenarios (Cao et al., 2023; Zhou et al., 2023; Chen et al., 2023; Alnegheimish et al., 2024).

Despite significant advancements, current LLM-based spatio-temporal models exhibit critical limitations. **First**, while these models are commonly applied for numerical tasks such as forecasting, their use in inferential problem-solving, such as reasoning or decision-making, remains underexplored (Li et al., 2024b; Zhou et al., 2023; Zhang et al., 2023; Yuan et al., 2024). For example, a forecasting system might violate real-world constraints, and erroneously predict traffic speeds that exceed safety thresholds, thus limiting interpretability and reliability of their outputs. **Second**, although foundation models generalize well, their performance compared to specialized smaller spatio-temporal models, is still debatable (Tan et al., 2024; Kambhampati et al., 2024). This raises the need to reassess the trade-offs between scalability, efficiency, and task optimization by proposing a hybrid approach that combines the strengths of foundation models and expert spatio-temporal models. **Third**, most

current models are restricted to fixed spatio-temporal input formats; typically tensors with pre-defined dimensions (e.g. [batch, time, location, feature]) and struggle with processing natural language queries, highlighting a gap in their utility as general-purpose AI systems (Zhou et al., 2023; Chen et al., 2023; Liu et al., 2024b; 2025).

Building on these limitations, few recent studies have explored models for spatio-temporal reasoning, though they predominantly focus on highly task-specific inference problems. Majority of these approaches convert spatio-temporal data into textual descriptions, for processing by LLMs (Peng et al., 2025; Chen et al., 2024; Guo et al., 2024). This translation often leads to significant information loss, shallow reasoning, and inability to capture complex dependencies inherent in spatio-temporal phenomena. To tackle these limitations, certain program-based approaches have been introduced. For example, UrbanLLM (Jiang et al., 2024) finetunes LLMs for urban planning by breaking down queries into sub-tasks handled by spatio-temporal AI models. However, its effectiveness is limited by the specific urban contexts it was trained on, reducing generalizability, and its reliance on pre-trained models may hinder detailed geospatial understanding. Similarly, TS-Reasoner (Ye et al., 2024) decomposes complex time-series tasks, yet it remains confined to niche time-series applications within climate and energy data, highlighting a gap in versatility and broader applicability. Moreover, the outputs of the aforementioned models are often brief and lack the depth required for long-form answers, limiting their practical utility. Long-form question answering (Fan et al., 2019) where comprehensive, explanatory, and interpretable outputs are generated from complex inputs still remains underexplored in spatio-temporal settings.

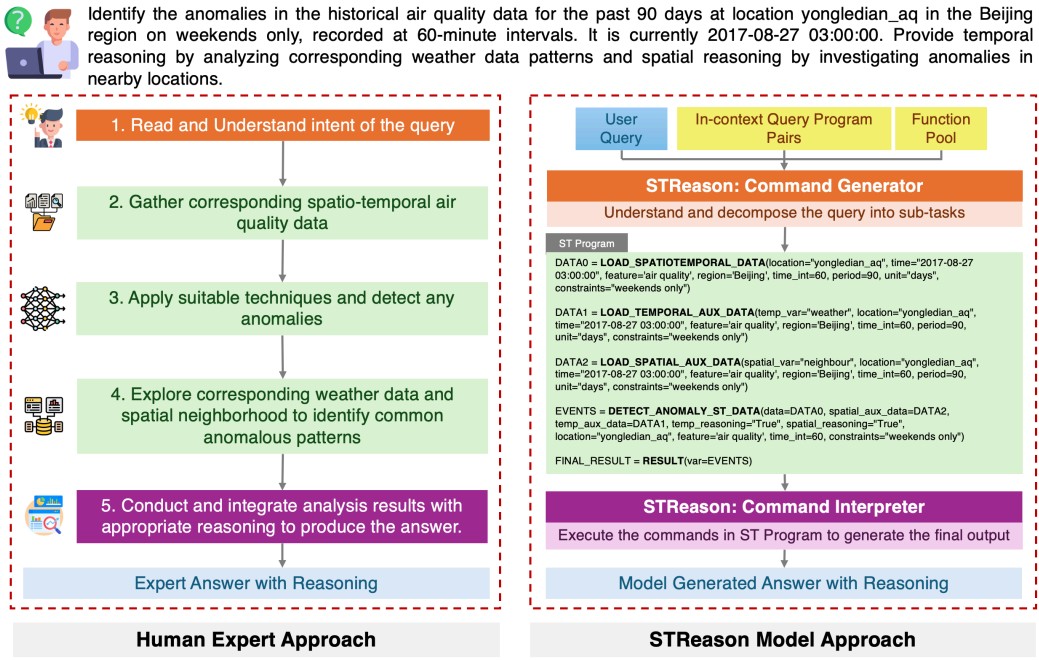

Figure 1: Comparison between Human Expert and STReason Model workflows for answering a complex spatio-temporal query.

Addressing these limitations requires a paradigm shift to a task and domain-agnostic framework that can adapt to complex spatio-temporal problems and generate outputs with rich context, depth, and clarity making results more interpretable and actionable. Consider answering a query on analyzing historical air quality data for anomalies, integrating temporal and spatial reasoning by considering corresponding weather patterns and nearby locations as shown in Figure 1. Typically, a human expert (Left) begins by understanding the intent of the query, followed by executing a series of analytical steps and finally providing a comprehensive answer that synthesizes all gathered insights with appropriate reasoning. While existing LLM-based systems can handle these individual steps in isolation, they fall short of effectively executing complex, language driven spatio-temporal queries end-to-end. They struggle to fully comprehend inputs, integrate multi-dimensional knowledge, and

produce reasoned, actionable outputs. Bridging this gap requires advanced LLM agents that function like full-stack data scientists, supporting the entire pipeline from data ingestion and analysis to interpretation and decision-making.

In response to these challenges, we introduce STReason, a novel framework that combines the reasoning and comprehension strengths of LLMs with the analytical power of state-of-the-art spatio-temporal models for multi-task inference and execution. Without requiring task-specific fine-tuning, STReason leverages LLMs to decompose complex tasks, articulated in natural language, into structured programs ("ST Program") using predefined in-context query–program example pairs and a Function Pool; a curated dictionary of available modules and their specifications that guides the LLM in aligning sub-tasks with the appropriate executable functions (see Figure 1 (Right)). The generated ST Programs are then executed by specialized, end-to-end trained models or tailored analytical programs to produce coherent, long-form answers that go beyond simple numeric predictions to include structured reasoning, explanatory narratives and meaningful interpretations inspired by long-form question answering (Fan et al., 2019). A demonstration of STReason can be viewed at: https://anon.to/T5lL94.

STReason revolutionizes spatio-temporal data analysis by addressing the limitations of conventional LLMs or LLM agents with limited spatio-temporal understanding (Manvi et al., 2024; Shen et al., 2023). Unlike task-specific models (Jiang et al., 2024; Li et al., 2024b), STReason requires no pre-training, enhancing flexibility and generalization across diverse domains. Highlighting its versatility, we apply STReason to three primary tasks in the traffic and air quality domains. These tasks include (i) Spatio-temporal Analysis, (ii) Spatio-temporal Anomaly Detection, and (iii) Spatio-temporal Prediction and Reasoning. The modular nature facilitates easy customization, promoting adaptation to new tasks without significant retraining. Moreover, STReason's structured execution allows for validation of logic and inspection of intermediate outputs crucial for high accountability domains like environmental monitoring. To enable rigorous evaluation, we also introduce a new benchmark dataset specifically designed for long form spatio-temporal reasoning. Unlike existing datasets that focus narrowly on single task, our dataset includes multi-task natural language queries, corresponding structured programs, and a ground-truth answer annotated with the key components expected in a comprehensive response. This dataset offers a valuable foundation for systematically evaluating any spatio-temporal reasoning model. We summarize our contributions as follows:

1. We propose STReason, a novel framework that decomposes complex multi-faceted spatio-temporal queries into executable steps and produce interpretable and well-reasoned outputs without human intervention.

2. We develop a benchmark dataset spanning three core tasks: Spatio-temporal Analysis, Anomaly Detection, and Prediction and Reasoning from real-world traffic and air quality data to rigorously assess any reasoning model, including STReason.

3. We develop a systematic evaluation framework to assess long-form reasoning responses for spatio-temporal queries, based on constraint adherence, factual accuracy, and logical coherence.

4. Extensive experiments demonstrate that STReason excels in spatio-temporal task execution and inference compared to advanced LLMs, highlighting their limitations in this domain.

## 2 RELATED WORK

**Large Language Models for Spatio-temporal Tasks** LLMs are driving significant advancements in spatio-temporal analysis (Jin et al., 2023b) through diverse tuning-based (Zhou et al., 2023; Liu et al., 2024b;c), prompt-based (Gruver et al., 2023; Zhang et al., 2023; Wang et al., 2023a), and foundation model approaches (Liang et al., 2024; 2025). While tuning-based methods risk catastrophic forgetting, and non-tuning methods rely on manual prompt engineering, task-specific foundation models face high development costs and limited generalization. Overall, these models are designed for fixed-format tensor inputs and scalar outputs, making them unsuitable for natural language queries, multi-task inference, or long-form reasoning. To address this gap, STReason introduces a task and domain agnostic reasoning framework along with a new benchmark dataset and evaluation protocol tailored for interpretable, multi-step spatio-temporal question answering, beyond what traditional metrics can capture.

**Reasoning with Foundation Models** Recent advances in foundation models (e.g., GPT-4o, DeepSeek-R1) extend reasoning capabilities beyond earlier versions (e.g., GPT-3), with improved context-awareness and domain adaptation. Techniques such as Chain-of-Thought (CoT) prompting and program-based methods improve logical deduction and generalize to visual, tabular, and time-series data (Yao et al., 2023; Wang et al., 2023b; Gupta & Kembhavi, 2023; Wang et al., 2024; Ye et al., 2024). The scope of reasoning has also broadened to include commonsense, numerical, and causal tasks by integrating contextual and quantitative signals (Li et al., 2024a;b; Zhang & Wan, 2023). Agent-based systems (JIAWEI et al., 2024; Gao et al., 2024) integrate tool use, while fine-tuned approaches (Kong et al., 2025) incur high costs and remain prone to hallucinations. These methods further lack alignment with spatio-temporal data and struggle to generalize beyond their training scope. In contrast, STReason is training-free and enables robust, constraint-aware, long-form reasoning across multiple spatio-temporal domains (see Appendix A.1).

## 3 METHODOLOGY

We introduce the STReason framework which operates in two primary stages: the command generation and command execution. Given a user query, in the first stage natural language queries are converted into structured commands by a Command Generator, which leverages the in-context learning abilities of LLMs. Using well-crafted query-program pairs along with a structured Function Pool of available interpreter modules, it generates a sequence of executable commands referred to as an "ST Program" tailored to each query. Each command in the ST Program activates one of the specific modules within our framework, which include state-of-the-art spatio-temporal prediction models, language processing units and data processing sub-routines. In the second stage, the generated ST Program is executed by a Command Interpreter that maps each command to its corresponding function, generating an integrated final response (see Figure 1 (Right)). This modular pipeline ensures that inputs are sequentially processed and integrated to deliver comprehensive, well-reasoned outputs.

### 3.1 COMMAND GENERATOR

The Command Generator is responsible for translating complex natural language queries into executable ST-Programs. Specifically, it decomposes complex spatio-temporal queries into manageable sub-tasks, leveraging the in-context learning capabilities of LLMs. To enhance grounding and reduce ambiguity in sub-task selection, STReason augments the input prompt with a Function Pool; a curated collection of function specifications that serves as a structured knowledge base for the model.

Each function in the pool includes its definition, input parameter details and output structures as shown in A.2. This structured reference not only clarifies available modules but also helps to resolve ambiguity during program generation, particularly when in-context examples lack alignment. For example, if a user query requests seasonality analysis but the provided examples only cover trend analysis, the Function Pool guides the LLM to select the correct function ("ANALYZE_SEASONALITY") by referencing its defined syntax and purpose.

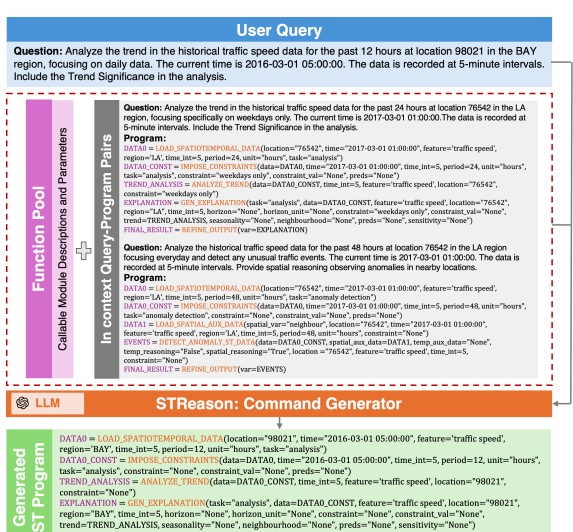

Figure 2: STReason Command Generator uses a Function Pool and in-context query-program pairs to generate executable ST-Program for a given user query.

It also improves generalization to diverse query phrasing. For example, a query like "Find anything unusual in pollution patterns over weekends" can still be correctly mapped to the anomaly detection function, despite differing wording from prior examples. Together with the input query–program pairs,

the Function Pool enables the Command Generator to produce a clear and interpretable program-based reasoning path, as illustrated in Figure 2. We use GPT-3.5 Turbo as the backbone LLM, due to the balance of its cost-efficiency and reasoning capability.

Each step of the generated program (i.e., ST-program) corresponds to a specific module within the framework, designed for analytical, predictive, or inference tasks. Particularly, each command line of an ST-program includes a module name (e.g., "ANALYZE_TREND", "DETECT_ANOMALY"), input arguments (e.g., "data", "location"), and an output variable (e.g., "EVENTS", "PREDICTION"). These naming conventions help the LLM understand and map inputs to the appropriate functions accurately. The final output from the command generator is a complete, interpretable program that defines all steps, inputs, and outputs needed to answer the query. Example ST-programs for different tasks are shown in Appendix A.3.

## 3.2 COMMAND INTERPRETER

The Command Interpreter sequentially executes the ST-Program, functioning like a traditional programming language interpreter. It accesses a library of pre-defined modules, invoking them as required by the program. It ensures that each command is executed in the correct order and that data flows correctly between steps, maintaining consistency and accuracy throughout the process. Finally, the results from each command are integrated to build a comprehensive response to the initial query, effectively turning raw data into insightful conclusions.

STReason features 12 specialized modules supporting three core spatio-temporal tasks: Analysis, Anomaly Detection and Prediction and Reasoning (see Figure 3). Inspired by VISPROG (Gupta & Kembhavi, 2023), each module is implemented as a Python class with methods for parsing inputs, executing computations, and summarizing outputs. Full module details are provided in Appendix A.4. This modular architecture simplifies the integration of new modules into STReason, requiring only the development and registration of a new module class. To further enhance transparency and user comprehension, each module gener-

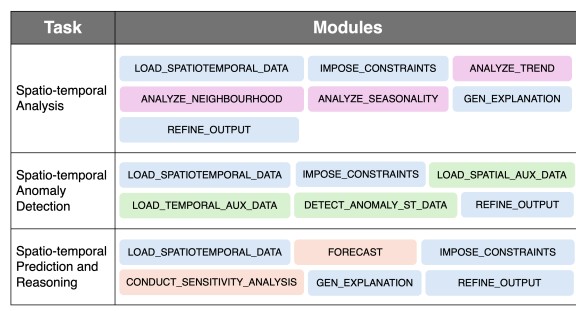

Figure 3: Command Interpreter Modules

ates a detailed textual summary of its operations, including inputs, processes, and outputs. The Command Interpreter compiles these into a complete execution rationale, offering clear insights into the program's logic and intermediate steps facilitating debugging and refinement (see Appendix A.5).

## 4 EXPERIMENTS

This section presents a comprehensive evaluation of STReason across key spatio-temporal tasks each requiring varying degrees of data interpretation, inference, and constraint-based reasoning. STReason is benchmarked against several advanced LLM-based baselines to assess its effectiveness in multi-task spatio-temporal inference and execution (Section 4.2). We further validate performance through a structured human evaluation that compares the quality and coherence of model-generated responses. (Section 4.3). Additionally, we perform an ablation study to analyze the influence of key factors on program generation accuracy (Section 4.4).

## 4.1 EXPERIMENTAL SETUP

**Dataset Creation** Due to the lack of standardized datasets for evaluating multi-task spatio-temporal reasoning models, we construct a new dataset tailored to evaluate such capabilities. It encompasses three representative tasks; Analysis, Anomaly Detection and Prediction and reasoning, each reflecting distinct spatio-temporal inference challenges. The dataset comprises 150 structured instances, each including a natural language query, a step-by-step program executable by STReason's Program Generator, and a corresponding ground truth answer. To ensure broad coverage and generalizability,

queries vary across regions, temporal intervals, forecast horizons, and domain-specific constraints. These instances are derived from four diverse real-world datasets including PEMS-BAY and METR-LA (Li et al., 2017) for traffic flow, and Beijing[1] and Shenzhen[2] for air quality. This dataset not only enables rigorous evaluation of the STReason framework but also aims to serve as a general-purpose benchmark for the broader research community.

**Baselines** We compare STReason framework against six advanced LLM baselines; LLaMA-2-7B, Vicuna-7B-v1.5, GPT-3.5 Turbo, GPT-4o Mini, GPT-4 and DeepSeek-V3. These baselines span a range of model sizes and architectural designs, allowing for a comprehensive evaluation across different reasoning capacities and generalization abilities. All models are accessed through public APIs, except for LLaMA-2-7B and Vicuna-7B-v1.5, which are run locally using open-source weights. Refer Appendix A.6 for baseline details and inference settings.

**Evaluation Metrics** To the best of our knowledge, no standardized metrics exist for systematically evaluating long-form question answering and reasoning in spatio-temporal tasks. To address this gap, we propose a novel evaluation framework that jointly measures the correctness, interpretability and reasoning quality of model-generated responses. Informed by the key principles of question answering and reasoning (Sun et al., 2023), three scores are defined as follows:

- **Constraint Adherence Score**: Measures whether the generated answer satisfies all query-specified constraints (e.g. temporal granularity, thresholds), using a binary scoring system.

- **Factuality Score**: Assess the presence and correctness of required analytical components (e.g. detected trends, anomaly timestamps, predicted values) by comparing them to a structured ground truth, scored as the ratio of correct components to expected components.

- **Coherence Score**: Assess the clarity and logical progression of the explanation on a 3-point scale, reflecting the overall coherence of the response.

In addition to evaluating overall model performance, we also conduct an in-depth assessment of the Program Generator as part of our ablation study (see Section 4.4). Here, the three metrics Precision, Recall, and F1 Score are computed, considering a program step correct if it matches in module type, input arguments, parameter values, and order. Full details of the evaluation procedure and corresponding prompt formulations are provided in Appendix A.7.

## 4.2 MAIN RESULTS

Table 1a presents a comparative evaluation of STReason against six advanced LLM baselines across three evaluation metrics. Accordingly, the results clearly demonstrate the superiority of STReason in effectively handling complex spatio-temporal tasks requiring both computational precision and interpretability. Specifically, STReason achieves a perfect constraint adherence score, satisfying all task-specific requirements across queries. While DeepSeek-V3 also shows strong performance in this aspect, STReason stands out by consistently meeting all requirements across every case. The most pronounced improvement is observed in the factuality score, where STReason achieves 84.44%, demonstrating its' ability to extract relevant analytical components and produce factually correct responses. With regard to the coherence score, along with models like GPT-3.5, GPT-4o, and DeepSeek, STReason delivers a top performance of 100%, maintaining complete logical consistency in its explanations. This reflects the inherent strengths of advanced LLMs in generating fluent and well-structured language. Furthermore, STReason's improvements in factuality and constraint adherence are statistically significant compared to majority of the baselines (see Appendix A.8).

Overall, STReason's strong performance across all dimensions highlights its reliable internal mechanisms and task relevant knowledge to generate coherent and factually accurate outputs. These results further establish STReason as a powerful framework for long-form, multi-task spatio-temporal reasoning tasks. Beyond quantitative metrics, we observed notable qualitative differences in model outputs. STReason consistently produced detailed, specific and statistically grounded analyses addressing the spatio-temporal queries. In contrast, DeepSeek often produced structured steps but lacked actual computed results. GPT4, GPT-4o-mini and GPT-3.5 Turbo provided only high-level methods without executing specific analyses. Responses from Vicuna-7B and LLaMA-2-7B were frequently vague,

---

[1]`https://dataverse.harvard.edu/dataverse/whw195009`
[2]`https://www.microsoft.com/en-us/research/project/urban-air`

Table 1: Quantitative Comparison of STReason against Baseline Models. Scores are reported as Mean ± Std. The bold and underlined font show the best and the second best result respectively.

| | (a) Overall Performance | | | (b) Forecasting Accuracy | |
|---|---|---|---|---|---|
| **Model** | **Constraint Score** | **Factuality Score** | **Coherence Score** | **MAE** | **RMSE** |
| Vicuna-7B-v1.5 | (55.3 ± 49.9)% | (11.6 ± 26.4)% | (71.6 ± 35.9)% | 43.1 ± 87.6 | 44.1 ± 87.5 |
| LLaMA-2-7B | (75.3 ± 43.3)% | (12.4 ± 23.8)% | (87.8 ± 25.2)% | 20.9 ± 23.2 | 22.4 ± 23.7 |
| GPT-4 | (67.3 ± 47.1)% | (25.1 ± 34.9)% | (99.1 ± 7.7)% | 52.0 ± 19.7 | 52.7 ± 19.3 |
| GPT-3.5 Turbo | (88.7 ± 31.8)% | (26.1 ± 34.1)% | (100 ± 0.0)% | 22.6 ± 27.1 | 23.3 ± 27.4 |
| GPT-4o Mini | (98.7 ± 11.5)% | (29.2 ± 38.3)% | (100 ± 0.0)% | 7.9 ± 12.3 | 8.8 ± 13.5 |
| DeepSeek-V3 | (97.3 ± 16.2)% | (32.8 ± 40.4)% | (100 ± 0.0)% | 11.4 ± 18.0 | 12.4 ± 18.7 |
| STReason | **(100.0 ± 0.0)%** | **(84.4 ± 26.9)%** | **(100 ± 0.0)%** | **7.6 ± 11.9** | **8.4 ± 12.8** |

unstructured, or contained hallucinations (Appendix A.10). These observations highlight STReason's strength in delivering comprehensive, precise, and actionable outputs for complex spatio-temporal tasks. Notably, STReason also generalizes well to unseen domains (Appendix A.11).

We further conducted a forecasting accuracy comparison using Mean Absolute Error (MAE) and Root Mean Squared Error (RMSE), the standard metrics in spatio-temporal forecasting (Table 1b). Unlike the baselines, which often produced incomplete or missing predictions, STReason consistently generated complete, correctly formatted outputs. For fair comparison, baseline predictions were post processed by zero padding if no predictions were generated and repeating the last available value to complete partial prediction sequences. As shown in Table 1b, STReason achieves the lowest MAE and RMSE, demonstrating superior forecasting accuracy. Moreover, these forecasting improvements are statistically significant against most baseline models, further confirming its superior quantitative reliability (see Appendix A.8). Although models like GPT-4o and DeepSeek showed competitive results, the post-processing adjustments could have unfairly favored these models by smoothing missing values, potentially inflating their apparent performance. Despite this, STReason still achieves the best accuracy, confirming its strength in delivering both accurate quantitative predictions and logically sound reasoning outputs while also maintaining practical efficiency (see Appendix A.9).

### 4.3 HUMAN EVALUATION EXPERIMENT

To further validate the performance of STReason beyond automated metrics, we conducted a rigorous human evaluation to assess the credibility, clarity, and reasoning quality of model generated answers. This complements our quantitative results with human judgment, thereby offering deeper insights into the effectiveness of STReason in real world contexts. We recruited 27 evaluators with domain relevant expertise, nearly half holding Ph.D.s and the majority specializing in Computing, Statistics, or Data Science, ensuring a technically competent cohort (see Appendix A.12). For this study, 18 spatio-temporal queries were curated spanning three core task types targeted by STReason. Each query was paired with two answers, one from STReason and one from a randomly selected baseline, with each baseline compared exactly three times. The order of answers was randomized to avoid positional bias. Evaluators were asked to choose the better response and provide open-ended feedback to explain their preferences. Full evaluation details are provided in Appendix A.12.

Across all 486 comparisons (27 evaluators × 18 questions), STReason was preferred in 74.1% of responses on average demonstrating strong user preference and credibility of generated outputs. As shown in Figure 4, STReason consistently outperformed all baselines. Vicuna-7B exhibited the lowest preference rate, followed by LLaMA-2-7B, GPT-3.5 Turbo and GPT-4 respectively. Even against top competitors like GPT-4o and DeepSeek, STReason maintained a clear margin of superiority. Notably, there is a consistent alignment between human evaluator preferences (Figure 4) and the automated factuality scores (Table 1a). The order of preference remains similar in both evaluations, which highlights the robustness and reliability of the proposed evaluation framework. Furthermore, as illustrated in Figure 5, the task wise preference rate further reveals STReason's robust performance across spatio-temporal task categories. Specifically it achieves 84.57% preference rate in prediction and reasoning task, 77.78% in analysis, and 59.88% in anomaly detection. These results suggest

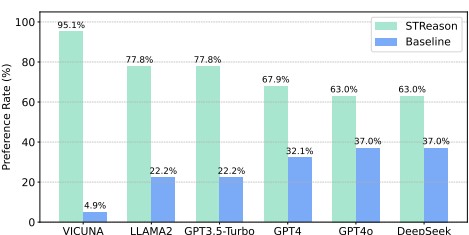 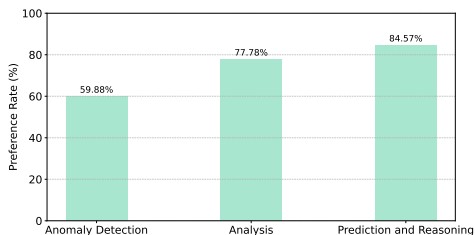

Figure 4: STReason vs. Baseline Preference Rate

Figure 5: STReason Task-wise Preference Rate

that while the model excels generally across all tasks, there's more room for improvement in tasks involving rare event detection.

Moreover, qualitative feedback from evaluators further supported these findings. As shown in Figure 6, the word cloud generated from open-ended responses where STReason was preferred, highlights attributes such as "detailed", "comprehensive", "clear", "structured", and "precise" reflecting STReason's strength in delivering well-organized, transparent, and informative answers. Overall, these outcomes affirm STReason's capacity to not only outperform baselines in constraint adherence and factual accuracy, but also to produce human preferred outputs that are coherent, more interpretable and contextually grounded.

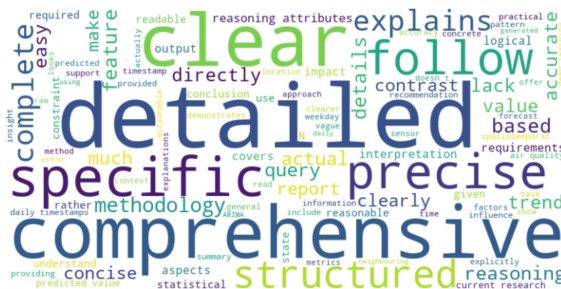

Figure 6: Human Evaluation: Qualitative Feedback

## 4.4 ABLATION STUDIES

To better understand the contribution of core components within the STReason framework, we conduct a series of ablation studies centered on the Command Generator. Each experiment is evaluated against ground-truth programs using Precision, Recall, and F1 Score as defined in Section 4.1.

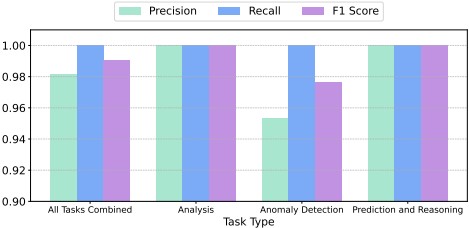 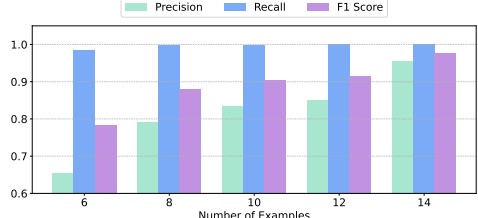

Figure 7: Effect of Task

Figure 8: Effect of No: of Incontext Examples

**Effect of Task Type:** We first assess performance across different task categories. As shown in Figure 7, the generator achieves perfect scores in both Analysis and Prediction tasks, while Anomaly Detection proves more challenging, showing a dip in precision, potentially due to the variability in task-specific steps and parameter configurations. However, when considering all tasks combined, the command generator maintains high overall accuracy, demonstrating strong generalizability.

**Effect of Number of In-Context Examples:** Next, we evaluate the impact of varying the number of in-context examples on program generation accuracy (see Figure 8). Precision and F1 Score steadily

improve as the number of examples increases from 6 to 14, while recall remains consistently high across all configurations. This trend highlights the importance of sufficient context in helping the model generate more accurate programs. Furthermore it is observed that, as the number of in-context examples increases, the proportion of task specific examples within the example pool rises from 14% with 6 examples to 21% with 14 examples contributing to the performance gains.

Table 2: Effect of In-context Example Variant

|  | Equal | Random | Test-query Include | Test-query Exclude |
|---|---|---|---|---|
| Precision | **0.9816** | 0.8327 | 0.9642 | 0.6091 |
| Recall | **1.0000** | 0.9973 | 1.0000 | 1.0000 |
| F1 Score | **0.9907** | 0.9026 | 0.9818 | 0.7571 |

**Effect of In-context Example Variant:** We then investigate how different in-context example variants influence program correctness. Four configurations were tested; a) *Equal Construction:* A fixed and balanced set of task-specific examples, b) *Random Construction:* A set of randomly selected examples from the example pool, c) *Test-Query Include:* A set of randomly selected examples including a certain percentage (20% in our case) of examples similar to the test query, c) *Test-Query Exclude:* A set of randomly selected examples excluding examples similar to the test query. As shown in Table 2, the Equal Construction yields the highest performance, affirming the importance of balanced, task-aligned examples. While the Test-Query Include setup also performs strongly, the Random setup demonstrates a noticeable drop in precision due to decreased example task alignment. The Test-Query Exclude configuration exhibits the lowest performance across metrics, highlighting the challenge of generating correct programs when the example pool lacks task relevance to the query.

Table 3: Effect of Function Pool

|  | Precision | | | | F1 Score | | | |
|---|---|---|---|---|---|---|---|---|
|  | Equal | Random | Test-Query Include | Test-query Exclude | Equal | Random | Test-Query Include | Test-Query Exclude |
| W/O Function Pool | 0.9816 | 0.8327 | 0.9642 | 0.6091 | 0.9907 | 0.9026 | 0.9818 | 0.7571 |
| With Function Pool | 0.9874 | 0.9678 | 0.9687 | 0.8440 | 0.9936 | 0.9836 | 0.9841 | 0.9132 |

**Effect of Function Pool:** We further evaluate the benefit of augmenting examples with a curated function pool. As shown in Table 3, the presence of the function pool significantly boosts precision and F1 scores across all example construction settings. It is particularly beneficial in the Test-Query Exclude setting, where query relevant examples are absent. These findings highlight the value of providing structured functional knowledge alongside in-context examples in improving program generation accuracy, especially when task-specific cues in the examples are limited.

## 5 CONCLUSION, LIMITATIONS AND FUTURE WORK

In this work, we introduced STReason, a novel framework for spatio-temporal multi-task inference and reasoning that seamlessly integrates large language models with state-of-the-art spatio-temporal models and analytical workflows. Through extensive benchmarking we show that, STReason outperforms advanced LLM baselines in spatio-temporal inference and execution tasks, achieving superior constraint adherence, factual accuracy, and reasoning coherence. Notably, our findings are further validated through structured human evaluations, reinforcing the credibility and robustness of the proposed automatic evaluation metrics. While effective, STReason currently relies on manually curated in-context examples which may limit scalability to unseen task types without further adaptation. Additionally, performance on tasks such as anomaly detection remains an area for improvement. Future work will focus on automating example retrieval and function pool maintenance to further enhance generalization, expanding task coverage and improving adaptability and accuracy through integration with emerging spatio-temporal foundation models. We also plan to extend STReason into a unified reasoning agent that can operate interactively across multimodal spatio-temporal environments.

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

# A    APPENDIX

## A.1    MORE RELATED WORK

**Program-based Reasoning** Among several recent reasoning approaches, Program-based reasoning has been explored in several domains, including visual question answering (VISPROG (Gupta & Kembhavi, 2023),ViperGPT (Surís et al., 2023)), tabular reasoning (Chain-of-Table (Wang et al., 2024)), multimodal task orchestration (HuggingGPT (Shen et al., 2023)), mathematical problem solving (Program-of-Thoughts (Chen et al., 2022), PAL (Gao et al., 2023)) and spatio-temporal QA for ubran planning (UrbanLLM (Jiang et al., 2024). Table 4 contrasts these representative frameworks with STReason, in terms of domain, input modality, output format and key innovation.

Table 4: Comparison of STReason with representative program-based reasoning frameworks.

| Method | Domain | Input Modality | Output Format | Key Innovation |
|---|---|---|---|---|
| VISPROG | Vision QA | Images | Short categorical text/ Edited Images | Compositional visual reasoning programs |
| ViperGPT | Vision QA | Images | Short categorical text/ Edited Images | Integrating code-generation models into vision |
| Chain-of-Table | Tabular reasoning | Structured tables | Short categorical text/ Result Table | Reasoning chains over evolving tables |
| HuggingGPT | General AI services | Multimodal (text, image, audio) | Task-specific responses | LLM routes tasks to expert models |
| Program-of-Thoughts | Math/text QA | Text | Numeric or symbolic answers | Program synthesis for mathematical reasoning |
| UrbanLLM | Spatio-temporal QA (Urban Planning) | Location and Time indexed urban data | Numeric Forecasts/Short answers | Domain-tuned LLMs for traffic/urban tasks |
| **STReason (ours)** | **Spatio-temporal QA (Multi-Domain)** | **Location and Time indexed data** | **Numeric analysis + Long-form Reasoning + Execution Rationale** | **Analytical, constraint-aware execution for interpretable, multi-domain ST reasoning** |

While these systems share the paradigm of decomposing natural language queries into structured programs, direct comparisons with STReason are not feasible due to fundamental differences in problem domain, query types, input and output modalities, and evaluation criteria. For example, VISPROG processes visual inputs and handles questions such as *"How many muffins can each kid have in the picture?"*, whereas STReason is designed for analytical spatio-temporal queries such as *"Perform a trend, seasonality, and neighborhood analysis of the historical traffic speed data for the past 90 days at location 402117 in the BAY region."*. Similarly, in terms of UrbanLLM, although it adapts program-based reasoning for spatio-temporal QA in urban planning tasks, its scope remains limited to specific urban contexts and short numeric predictions, falling short of the broader analytical reasoning that STReason provides.

To further highlight why comparisons with existing program-based methods are infeasible, we tested representative STReason queries on several open-source program-based frameworks. Chain-of-Table returned *"Final answer: N/A"*, Program-of-Thoughts failed with *"Empty solver() function and*

*Prediction: None"*, and PAL (Program-aided Language Models) returned *"No results was produced. A common reason is that the generated code snippet is not valid or did not return any results."*. These outcomes show that, although effective in their respective domains, existing models are unable to process the structured spatio-temporal queries that STReason is designed to address. Moreover, as shown in Table 4, most existing program-based approaches typically output short categorical or numerical responses, while STReason generates long-form, interpretable outputs that integrate numerical analysis, constraint checks, and explanatory narratives. STReason therefore advances beyond both general program-based frameworks and domain-specific approaches such as UrbanLLM, introducing mechanisms for structured, actionable reasoning in spatio-temporal decision-making. In summary, STReason is a deliberate extension of program-guided reasoning to real-world problems requiring structured inputs, statistical analysis, and interpretable outputs, thereby filling a critical gap between LLMs and decision-support tools for spatio-temporal domains.

## A.2 FUNCTION POOL

The Function Pool serves as an explicit grounding mechanism for in-context learning within STReason. It contains structured descriptions of all callable modules, including their syntax, parameter definitions, and functional purpose. During program generation, this information is appended to the prompt, enhancing its ability to align sub-tasks with appropriate functions. This design helps ensure correct function usage, particularly in cases where the provided in-context examples are insufficient (Table 3). The structure of a sample function is shown in Figure 9.

---

**LOAD_SPATIOTEMPORAL_DATA**

**Call Signature:**
`LOAD_SPATIOTEMPORAL_DATA(location, time, feature, region, time_int, period, unit, task)`

**Description:**
Loads spatio-temporal data for a specific location and time period.

**Parameters:**
- `location (str)` — The geographical location identifier.
- `time (datetime)` — The current time.
- `feature (str)` — Feature of interest (e.g., `'traffic speed'`, `'air quality'`).
- `region (str)` — The broader geographical area.
- `time_int (int)` — Interval in minutes between data points.
- `period (int)` — Duration for which data is loaded.
- `unit (str)` — The time unit of the period (e.g., `'hours'`, `'days'`).
- `task (str)` — Task of the query (e.g., `'analysis'`, `'anomaly detection'`, `'prediction'`).

**Returns:**
`DataFrame` containing spatio-temporal data.

---

Figure 9: Structure of Sample Function

## A.3 TASK-WISE PROGRAM GENERATION

We showcase sample ST-program structures generated for a range of spatio-temporal tasks including Analysis 10, Anomaly Detection 11, and Prediction and reasoning 12.

**Seasonality Analysis**

**Question:** Perform a seasonality analysis of the historical air quality data for the past 50 days at location tianjin_aq in the Shenzhen region focusing specifically on weekends only. The current time is 2020-09-05 02:05:00.The data is recorded at 60-minute intervals. Analyze the Daily Seasonality Strength and Weekly Seasonality Strength.

**Program:**
```
DATA0 = LOAD_SPATIOTEMPORAL_DATA(location="tianjin_aq", time="2020-09-05 02:05:00", feature="air quality", region='Shenzhen', time_int=60, period=50, unit = "days",
task="analysis")
DATA0_CONST = IMPOSE_CONSTRAINTS(data=DATA0, time="2020-09-05 02:05:00", time_int=60, period=50, unit = "days", task="analysis", constraint="weekends only",
constraint_val="None", preds="None")
SEASONALITY_ANALYSIS = ANALYZE_SEASONALITY(data=DATA0_CONST, time_int=60, constraint="weekends only")
EXPLANATION = GEN_EXPLANATION(task="analysis", data=DATA0_CONST, feature='air quality', location="tianjin_aq", region="Shenzhen", time_int=60, horizon="None",
horizon_unit="None", constraint="weekends only", constraint_val="None", trend="None", seasonality="SEASONALITY_ANALYSIS", neighbourhood="None", preds="None",
sensitivity="None")
FINAL_RESULT = REFINE_OUTPUT(var=EXPLANATION)
```

**Trend Analysis**

**Question:** Analyze the trend in the historical traffic speed data for the past 12 hours at location 98021 in the BAY region, focusing on daily data. The current time is 2016-03-01 05:00:00. The data is recorded at 5-minute intervals. Analyze the trend significance.

**Program:**
```
DATA0 = LOAD_SPATIOTEMPORAL_DATA(location="98021", time="2016-03-01 05:00:00", feature='traffic speed', region='BAY', time_int=5, period=12, unit="hours",
task="analysis")
DATA0_CONST = IMPOSE_CONSTRAINTS(data=DATA0, time="2016-03-01 05:00:00", time_int=5, period=12, unit="hours", task="analysis", constraint="None",
constraint_val="None", preds="None")
TREND_ANALYSIS = ANALYZE_TREND(data=DATA0_CONST, time_int=5, feature='traffic speed', location="98021", constraint="None")
EXPLANATION = GEN_EXPLANATION(task="analysis", data=DATA0_CONST, feature='traffic speed', location="98021", region="BAY", time_int=5, horizon="None",
horizon_unit="None", constraint="None", constraint_val="None", trend=TREND_ANALYSIS, seasonality="None", neighbourhood="None", preds="None", sensitivity="None")
FINAL_RESULT = REFINE_OUTPUT(var=EXPLANATION)
```

**Neighbourhood Analysis**

**Question:** Conduct a neighbourhood analysis of the historical air quality data for the past 21 days at location dongsi_aq in the Beijing region, focusing specifically on weekends only. The current time is 2017-09-05 02:05:00. The data is recorded at 60-minute intervals. Analyze the spatial Neighbours.

**Program:**
```
DATA0 = LOAD_SPATIOTEMPORAL_DATA(location="dongsi_aq", time="2017-09-05 02:05:00", feature="air quality", region='Beijing', time_int=60, period=21, unit="days",
task="analysis")
DATA0_CONST = IMPOSE_CONSTRAINTS(data=DATA0, time="2017-09-05 02:05:00", time_int=60, period=21, unit="days", task="analysis", constraint="weekends only",
constraint_val="None", preds="None")
NEIGHBOURHOOD_ANALYSIS = ANALYZE_NEIGHBOURHOOD(feature='air quality', location="dongsi_aq", region='Beijing')
EXPLANATION = GEN_EXPLANATION(task="analysis", data=DATA0_CONST, feature='air quality', location="dongsi_aq", region="Beijing", time_int=60, horizon="None",
horizon_unit="None", constraint="weekends only", constraint_val="None", trend="None", seasonality="None", neighbourhood=NEIGHBOURHOOD_ANALYSIS, preds="None",
sensitivity="None")
FINAL_RESULT = REFINE_OUTPUT(var=EXPLANATION)
```

Figure 10: ST-program for Analysis Task

**Anomaly Detection**

**Question:** Analyze the historical air quality data for the past 7 days at location 4007 in the Shenzhen region focusing everyday and detect any unusual air quality patterns. The current time is 2015-04-06 06:00:00. The data is recorded at 60-minute intervals. Provide temporal reasoning observing patterns of corresponding weather data and spatial reasoning observing anomalies in nearby locations.

**Program:**
```
DATA0 = LOAD_SPATIOTEMPORAL_DATA(location="4007", time="2015-04-06 06:00:00", feature='air quality', region='Shenzhen', time_int=60, period=7, unit="days",
task="anomaly detection")
DATA0_CONST = IMPOSE_CONSTRAINTS(data=DATA0, time="2015-04-06 06:00:00", time_int=60, period=7, unit="days", task="anomaly detection", constraint="None",
constraint_val="None", preds="None")
DATA1 = LOAD_TEMPORAL_AUX_DATA(temp_var="weather", location="4007", time="2015-04-06 06:00:00", feature = 'air quality', region='Shenzhen', time_int=60, period=7,
unit="days", constraint="None")
DATA2 = LOAD_SPATIAL_AUX_DATA(spatial_var="neighbour", location="4007", time="2015-04-06 06:00:00", feature='air quality', region='Shenzhen', time_int=60, period=7,
unit="days", constraint="None")
EVENTS = DETECT_ANOMALY_ST_DATA(data=DATA0_CONST, spatial_aux_data=DATA2, temp_aux_data=DATA1, temp_reasoning="True", spatial_reasoning="True",
location="4007", feature='air quality', time_int=60, constraint="None")
FINAL_RESULT = REFINE_OUTPUT(var=EVENTS)
```

Figure 11: ST-program for Anomaly Detection Task

**Prediction and Reasoning**

**Question:** The current time is 2012-03-01 01:00:00. Predict the traffic speed of sensor ID 767542 in the LA region for the next 1 hour using the historical data of the past 1 hour with data points recorded at 5-minute intervals. Ensure the predicted traffic speed does not exceed the 100 km/h safety threshold. Analyze how daily timestamps and neighbouring sensors influence the traffic speed predictions.

**Program:**
```
DATA0 = LOAD_SPATIOTEMPORAL_DATA(location="767542", time="2012-03-01 01:00:00", feature='traffic speed', region="LA", time_int=5, period=1, unit="hours",
task="prediction")
PREDICTION = FORECAST(data=DATA0, location="767542", time="2012-03-01 01:00:00", feature='traffic speed', region="LA", time_int=5, period=1, unit="hours",
horizon=1, horizon_unit="hours")
PREDICTION_CONST = IMPOSE_CONSTRAINTS(data=DATA0, time="2012-03-01 01:00:00", time_int=5, period=1, unit="hours", task="prediction", constraint="traffic speed
threshold", constraint_val=100, preds=PREDICTION)
SPATIOTEMPORAL_SENSITIVITY = CONDUCT_SENSITIVITY_ANALYSIS(data=DATA0, preds=PREDICTION_CONST, location="767542", time="2012-03-01 01:00:00",
feature='traffic speed', region="LA", time_int=5, period=1, unit="hours", horizon=1, horizon_unit="hours")
EXPLANATION = GEN_EXPLANATION(task="prediction", data=DATA0, feature='traffic speed', location="767542", region="LA", time_int=5, horizon=1, horizon_unit="hours",
constraint="traffic speed threshold", constraint_val=100, trend="None", seasonality="None", neighbourhood="None", preds=PREDICTION_CONST,
sensitivity=SPATIOTEMPORAL_SENSITIVITY)
FINAL_RESULT = REFINE_OUTPUT(var=EXPLANATION)
```

Figure 12: ST-program for Prediction and Reasoning Task

## A.4 COMMAND INTERPRETER MODULE DETAILS

We illustrate the details of the 12 modules within the STReason framework below to better understand their functionalities and specifications.

---

**LOAD_SPATIOTEMPORAL_DATA**

**Call Signature:**
LOAD_SPATIOTEMPORAL_DATA(location, time, feature, region, time_int, period, unit, task)
**Description:**
Loads spatio-temporal data for a specific location and time period. Returns a DataFrame containing the relevant data based on parameters such as location, feature, and time interval.

---

**LOAD_SPATIAL_AUX_DATA**

**Call Signature:**
LOAD_SPATIAL_AUX_DATA(spatial_var, location, time, feature, region, time_int, period, unit, constraint)
**Description:**
Loads auxiliary spatial data such as neighbourhood data to support spatial reasoning in tasks.

---

**LOAD_TEMPORAL_AUX_DATA**

**Call Signature:**
LOAD_TEMPORAL_AUX_DATA(temp_var, location, time, feature, region, time_int, period, unit, constraint)
**Description:**
Loads auxiliary temporal data such as weather data to support temporal reasoning in tasks.

---

**IMPOSE_CONSTRAINTS**

**Call Signature:**
IMPOSE_CONSTRAINTS(data, time, time_int, period, unit, task, constraint, constraint_val, preds)
**Description:**
Applies data constraints relevant to analysis, prediction, or anomaly detection. It can filter the data or enforce threshold-based rules for predictive outputs.

---

**ANALYZE_TREND**

**Call Signature:**
ANALYZE_TREND(data, feature, location, time_int, constraint, output_var)
**Description:**
Performs trend detection on the selected feature. Returns a text summary of the trend analysis.

---

**ANALYZE_SEASONALITY**

**Call Signature:**
ANALYZE_SEASONALITY(data, time_int, constraint)
**Description:**
Analyzes seasonality patterns in the spatio-temporal data. Returns a text summary of the seasonality analysis.

---

## ANALYZE_NEIGHBOURHOOD

**Call Signature:**
`ANALYZE_NEIGHBOURHOOD(feature, location, region)`
**Description:**
Examines the spatial surroundings of a location to analyze feature behavior in neighboring areas. Often used to detect localized anomalies or support spatial reasoning.

## GEN_EXPLANATION

**Call Signature:**
`GEN_EXPLANATION(task, data, feature, location, region, time_int, horizon, horizon_unit, constraint, constraint_val, trend, seasonality, neighbourhood, preds, sensitivity)`
**Description:**
Generates a comprehensive narrative explaining the results based on trend, seasonality, neighborhood context, predictions, constraints, and sensitivity analysis.

## DETECT_ANOMALY_ST_DATA

**Call Signature:**
`DETECT_ANOMALY_ST_DATA(data, spatial_aux_data, temp_aux_data, temp_reasoning, spatial_reasoning, location, feature, time_int, constraint)`
**Description:**
Identifies anomalies using both core and auxiliary data sources.

## FORECAST

**Call Signature:**
`FORECAST(data, location, time, feature, region, time_int, period, unit, horizon, horizon_unit)`
**Description:**
Performs forecasting on the selected feature based on historical data.

## CONDUCT_SENSITIVITY_ANALYSIS

**Call Signature:**
`CONDUCT_SENSITIVITY_ANALYSIS(data, preds, location, time, feature, region, time_int, period, unit, horizon, horizon_unit)`
**Description:**
Analyzes how changes in input data influence the predictions, offering insights into both spatial and temporal sensitivity for more robust interpretations.

## REFINE_OUTPUT

**Call Signature:**
`REFINE_OUTPUT(var)`
**Description:**
Outputs the final result from any task in a standardized format. This can be a summary string, table, or numeric result depending on the task context.

## A.5 Execution Rationale

To demonstrate the transparency and traceability of the STReason framework, we include below the execution rationale for two queries. These rationales are automatically generated by the Command Interpreter during the program execution stage.

***Spatio-temporal Analysis Query:*** *Perform a trend, seasonality, and neighbourhood analysis of the historical traffic speed data for the past 90 days at location 402117 in the BAY region, focusing on weekdays only. Analyze the Trend Significance, Daily Seasonality Strength, Weekly Seasonality Strength, and Neighbours. The data is recorded at 5-minute intervals and the current time is 2017-03-04 01:40:00.*

```
Loaded data for Location: 402117, Feature: traffic speed, Time Range: From 2016-12-04 01:40:00 to 2017-03-04 01:40:00.
Imposed constraints and retrieved data from weekdays only.
Trend Analysis Conducted.
Trend plot saved at ['./Visualizations/402117_traffic speed_trend_plot.png'].
Seasonality Analysis Conducted.
Neighbourhood Analysis Conducted.
Neighbouring locations detected:[['402056', '402057', '402058', '402118']].
Final Explanation Generated.
```

Figure 13: Execution Rationale for Spatio-temporal Analysis Query

***Spatio-temporal Prediction Query:*** *What will be the traffic speed at location 409524 in the BAY region for the next 35 minutes, based on historical data from the past 1 hour recorded at 5-minute intervals? The current time is 2017-02-14 03:00:00. Please ensure that the predicted traffic speed does not exceed 100 km/h. Also, analyze how daily timestamps and neighbouring sensors impact the accuracy of traffic speed predictions.*

```
Loaded data for Feature: traffic speed, Time Range: From 2017-02-14 02:00:00 to 2017-02-14 03:00:00.
Forecasted traffic speed for location 409524 for next 35 minutes using Graph Wavenet Model.
Imposed constraints considering traffic speed threshold of 100.
Spatial and Temporal Sensitivity analysis conducted.
Final Explanation Generated.
```

Figure 14: Execution Rationale for Spatio-temporal Prediction Query

## A.6 Baseline Model Details

**Model Descriptions**

- **LLaMA-2-7B**[3]**:** An open-source LLM developed by Meta AI with 7 billion parameters.
- **Vicuna-7B-v1.5**[4]**:** A fine-tuned variant of LLaMA-2-7B optimized for dialogue and instruction-following tasks, supporting up to 16k context length.
- **GPT-3.5 Turbo** (Ye et al., 2023): A widely used commercial LLM by OpenAI, optimized for speed and cost-effectiveness while retaining strong reasoning capabilities.
- **GPT-4o Mini** [5]: A lightweight, high-performance version of GPT-4o with improved efficiency. It supports multimodal inputs and enhanced contextual reasoning in real-time applications.
- **GPT-4** (Achiam et al., 2023): OpenAI's flagship model known for its robust generalization and reasoning performance across a wide range of tasks, including multi-step and constraint-based inference.
- **DeepSeek-V3** (Liu et al., 2024a): A recent open-weight LLM developed by DeepSeek, trained on an extensive web-scale corpus with strong performance on benchmark reasoning and coding tasks.

---

[3]https://huggingface.co/meta-llama/Llama-2-7b-chat-hf
[4]https://huggingface.co/lmsys/vicuna-7b-v1.5
[5]https://platform.openai.com/docs/models/gpt-4o-mini

**Inference Settings** To ensure a fair comparison across models, we standardized the inference settings for all LLM baselines. For open source models LLaMA-2-7B and Vicuna-7B-v1.5 we used HuggingFace Transformers and answer generation was performed with `temperature=0.7` and `max_new_tokens=4096`. These two models were run locally using HuggingFace Transformers on a single NVIDIA A40 GPU (46GB memory, CUDA 12.4). The remaining models including GPT-3.5 Turbo, GPT-4o Mini, GPT-4, and DeepSeek-V3 were accessed via their respective public APIs. For GPT-based models, we used the ChatCompletion endpoint with consistent parameters including `temperature=0.7`, `top_p=1`, and `max_tokens=4096`. DeepSeek-V3 was queried using `temperature=1.3` which is used for general conversation and default decoding parameters. This uniform setup ensured consistent evaluation conditions across all models. Table 5 provides a summary of further details of baseline models.

Table 5: Summary of LLM Baselines used for Comparison.

| Model | Parameter Size | Context Length | Access Type |
|---|---|---|---|
| LLaMA-2-7B | 7B | 4k tokens | Open-source (HuggingFace) |
| Vicuna-7B-v1.5 | 7B | 16k tokens | Open-source (HuggingFace) |
| GPT-3.5 Turbo | 175B (est.) | 16k tokens | API (OpenAI) |
| GPT-4o Mini | Unknown | 128k tokens | API (OpenAI) |
| GPT-4 | 1.7T (est.) | 8k tokens | API (OpenAI) |
| DeepSeek-V3 | 671B | 128k tokens | API (OpenAI) |

**Prompting Strategy for Baseline Models**

To ensure fair comparison, each baseline model received the same input query and data as STReason. Additionally, a task-specific system prompt was prepended to guide the model's behavior (e.g., "You are an expert in spatio-temporal forecasting.). This format was adapted across all tasks and baseline models and input data was passed as structured tables. Given below is a representative example of a prediction task prompt.

---

**Example Prompt for Prediction Task**

You are an expert in spatio-temporal forecasting. The current time is `2017-01-04 07:00:00`. Predict the air quality of location `dongsi_aq` in the Beijing region for the next 6 hours using the historical data of the past 24 hours with data points recorded at 60-minute intervals. Ensure the predicted air quality does not exceed the 75 $\mu g/m^3$ safety threshold.
Input Data:

| Timestamp | dongsi_aq ($\mu g/m^3$) |
|---|---|
| 2017-01-03 08:00:00 | 56.0 |
| 2017-01-03 09:00:00 | 60.0 |
| 2017-01-03 10:00:00 | 62.0 |
| 2017-01-03 11:00:00 | 64.0 |
| ⋮ | ⋮ |
| 2017-01-04 07:00:00 | 68.0 |

---

## A.7 EVALUATION METRICS

We propose a novel evaluation framework that jointly assesses the correctness, interpretability, and reasoning quality of model-generated responses, using three distinct metrics described below:

**Constraint Adherence Score:** This binary metric assesses whether the generated response satisfies all explicit constraints in the query (e.g., thresholds, time spans). A structured prompt (see Figure 15) is used to check constraint fulfillment through an LLM-based verifier, returning 'True' or 'False'. Finally, the scores are averaged over all queries.

**Factuality Score:** This metric evaluates the correctness and completeness of key analytical components (e.g., trend values, anomalies, predictions) against the ground truth. As shown in Figure 16, a

Consider the following Question, Constraint, and Answer:

**Question**   : *[Query]*
**Constraint** : *[Associated Constraint]*
**Answer**     : *[Model generated Answer]*

Conduct the following analysis:
**Step 1:** Assess if the answer aligns with the constraint:
 - [Constraint] Is met/not met because [reason based on the answer part].

**Step 2:** Summarize the findings:
 - Constraint is [met/not met].

**Final Answer:** Respond 'True' if the constraint is met, respond 'False' otherwise.

Figure 15: Prompt for assessing Constraint Adherence

prompt guides the LLM to extract and validate components. The score is computed as the proportion of correct components identified. The final score is the average rating across all queries.

Consider the following details to verify if specific 'Ground Truth Answer Components' with defined values from 'Ground Truth Answer' are accurately reflected in the 'Model Generated Answer'.

**Ground Truth Answer Components :** *[Ground Truth Answer Components]*
**Ground Truth Answer**                 : *[Ground Truth Answer]*
**Model Generated Answer**              : *[Model Generated Answer]*

Conduct the following analysis:

**Step 1:** Component Presence Check: For each component in 'Ground Truth Answer Components', check if it is explicitly mentioned in the 'Model generated answer' with the values or details as given in 'Ground Truth Answer'.
Record 'present with correct details' for components found with matching details/values and 'absent or incorrect details' for those that are not present or have incorrect details.

**Step 2:** Correct number of components: Count the number of components identified as 'present with correct details'.

**Final Answer:** Respond with the Correct number of components. This response should be a numeric value only.

Figure 16: Prompt for assessing Factual Correctness

**Coherence Score:** This metric evaluates the logical consistency and clarity of model generated answer. An LLM evaluator (Figure 17) rates model generated answers on a 3-point ordinal scale based on transition quality and reasoning flow. The final score is the average rating across all samples.

We also assess the accuracy of the ST-Program generated by the Command Generator, which form the backbone of STReason's execution process. A program step is considered correct if it matches the reference in module type, argument names, parameter values, and order. To measure program generation performance, we adopt the following metrics:

> **Question** : *[Query]*
> **Answer** : *[Model generated Answer]*
>
> Evaluate the answer for the given question:
>
> **Step 1:** Coherence Evaluation: Are the transitions between points in the explanation smooth and logical?
> - Cohesion score = 1: Assign if transitions are abrupt or disjointed.
> - Cohesion score = 2: Assign if transitions are generally clear with minor issues.
> - Cohesion score = 3: Assign if the explanation flows seamlessly and logically.
>
> **Step 2:** Format the evaluation as follows:
> - **Cohesion score** = <Score>
> - Explanation: <Explanation for assignment of specific scores>

Figure 17: Prompt for assessing Logical Coherence

**Precision** evaluates the accuracy of predicted steps by measuring the proportion of generated commands that are both syntactically and semantically correct.

$$\text{Precision} = \frac{\text{True Positives (TP)}}{\text{True Positives (TP)} + \text{False Positives (FP)}} \tag{1}$$

**Recall** measures the completeness of the generated program by calculating the proportion of required steps that the model successfully included.

$$\text{Recall} = \frac{\text{True Positives (TP)}}{\text{True Positives (TP)} + \text{False Negatives (FN)}} \tag{2}$$

**F1-score** provides a balanced measure by computing the harmonic mean of precision and recall, offering a single metric to capture the trade-off between completeness and correctness:

$$\text{F1 Score} = 2 \times \left( \frac{\text{Precision} \times \text{Recall}}{\text{Precision} + \text{Recall}} \right) \tag{3}$$

- True Positives (TP): Program steps that exactly match the ground truth in module type, argument names, and values.

- False Positives (FP): Steps generated by the model that either do not appear in the ground truth, or are incorrect in terms of function, parameters, or sequence.

- False Negatives (FN): Steps present in the ground truth but missing from the generated program, such as omitted data loading or post-processing functions.

A.8 STATISTICAL SIGNIFICANCE TESTING

To assess the robustness of STReason's performance improvements, we conducted paired Wilcoxon signed-rank tests over 150 test examples, comparing STReason against each baseline across all evaluation metrics.

**Overall Performance Significance:** The p-values reported in Table 6a indicate that STReason significantly outperforms all baselines in constraint adherence ($p < 0.05$), except GPT-4o Mini, which achieved comparable constraint scores. In terms of factuality, STReason outperformed all models with strong significance ($p < 0.0001$), confirming its reliability in producing accurate, data-grounded responses. For coherence, STReason achieved perfect scores along with GPT-3.5 Turbo, GPT-4o Mini, and DeepSeek, leading to tied outcomes and non-applicable tests. However, its advantage over LLaMA, and Vicuna are statistically significant. These results collectively validate the robustness of STReason across interpretability, factual soundness, and analytical precision.

**Forecasting Accuracy Significance:** Based on the p-values for forecasting accuracy metrics in Table 6b, STReason demonstrated statistically significant improvements in MAE and RMSE over

Table 6: Wilcoxon Signed-Rank Test p-values for STReason vs. Baselines

(a) Overall Performance | | | | (b) Forecasting Accuracy

| Model | Constraint Score | Factuality Score | Coherence Score | MAE | RMSE |
|---|---|---|---|---|---|
| Vicuna-7B-v1.5 | <**0.0001** | <**0.0001** | <**0.0001** | <**0.0001** | <**0.0001** |
| LLaMA-2-7B | <**0.0001** | <**0.0001** | <**0.0001** | **0.0005** | **0.0002** |
| GPT-4 | <**0.0001** | <**0.0001** | 0.1573 | <**0.0001** | <**0.0001** |
| GPT-3.5 Turbo | <**0.0001** | <**0.0001** | — | **0.0061** | **0.0028** |
| GPT-4o Mini | 0.1573 | <**0.0001** | — | 0.6652 | 0.3895 |
| DeepSeek-V3 | **0.0455** | <**0.0001** | — | 0.2268 | 0.1478 |

GPT-3.5 Turbo, GPT-4, LLaMA2, and Vicuna. The differences were not statistically significant compared to GPT-4o Mini and DeepSeek, likely due to similar performance and the effect of sequence post-processing. These results reinforce STReason's robustness in forecasting accuracy across diverse baselines.

## A.9 LATENCY AND COST ANALYSIS

To evaluate the practical feasibility of deploying STReason in real world decision-making settings, we further analyzed its latency, token usage and API cost across representative query types. All evaluations were conducted using GPT-3.5 Turbo on standard hardware without GPU acceleration.

Table 7: Average Latency by Task Type (in seconds)

| Task Type | Program Gen | Answer Gen | Total Time |
|---|---|---|---|
| Analysis | 4.27 | 5.04 | 9.31 |
| Anomaly Detection and Reasoning | 4.20 | 6.92 | 11.12 |
| Prediction and Reasoning | 4.59 | 10.91 | 15.5 |

**Latency Analysis:** Table 7 reports the average latency results for three task categories. Each query goes through two LLM calls, one for Program Generation and one for Answer Generation, including function execution and explanation synthesis. Analysis tasks are completed in approximately 9 seconds, demonstrating rapid end-to-end execution. Anomaly detection, which requires deeper contextual reasoning, takes slightly longer. Prediction tasks involve generating multi-step predictions and explanatory narratives, resulting in the highest average latency of approximately 16 seconds. Overall, all task types remain within an acceptable response window for interactive applications, confirming the practical feasibility of deploying STReason in urban monitoring and planning systems.

**Cost Analysis:** Token usage in STReason varies with the complexity and task type of the query. On average, queries are $\sim 80$ tokens, and responses are $\sim 364$ tokens. In addition, the input includes a structured Function Pool ($\sim 2211$ tokens) and in-context examples ($\sim 700$ tokens). This results in a total input size of $\sim (2800 - 3000)$ tokens per query, which remains well within the 16k context limit of GPT-3.5 Turbo. At OpenAI's current pricing for GPT-3.5 Turbo ($0.0015 per 1K input tokens and $0.002 per 1K output tokens), the estimated cost per query is approximately $0.0051, making STReason highly cost-efficient for scalable deployment.

## A.10 QUALITATIVE OUTPUT COMPARISON

To complement the quantitative metrics, we provide the following examples that illustrate commonly observed issues in baseline models. These examples highlight qualitative gaps such as hallucinations, vague or unstructured responses, and missing computations, contrasted with STReason's consistent, specific, and statistically grounded analyses of spatiotemporal queries.

---

**Example 1: Hallucination (Vicuna-7B-v1.5)**

**Query:** Perform a seasonality and neighbourhood analysis using the historical air quality data for the past 30 days at location yongledian_aq in the Beijing region, based on daily patterns. The data is recorded at 60-minute intervals. The current time is 2017-08-15 09:00:00.

**Generated Answer:**

1. Seasonality: The air quality data shows a clear seasonality pattern with higher values in the summer months and lower values in the winter months. Specifically, the average air quality in the summer months (June, July, and August) is significantly higher than the average air quality in the winter months (December, January, and February). This seasonality pattern is ...

---

**Issue:** The model hallucinates seasonal trends across summer (June–August) and winter (December–February), despite being provided with only 30 days of hourly data.

---

**Example 2: Vague, Unclear and Incomplete Response (Vicuna-7B-v1.5)**

**Query:** Examine the historical air quality data for the past 24 hours at location xizhimen-bei_aq in the Beijing region on weekdays only, with recordings at 60-minute intervals. The current time is 2017-04-29 09:00:00. Identify any anomalies and provide temporal reasoning by analyzing anomalies in corresponding weather data patterns and spatial reasoning by identifying anomalies in nearby locations.

**Generated Answer:**

A: Identify any anomalies in the data by considering temporal patterns in corresponding weather data and spatial anomalies in nearby locations.
B: Identify any spatial anomalies in the data by considering the spatial patterns in the weather data and the data itself.
C: Identify any temporal patterns in the data by considering the temporal patterns in the weather data and the data itself.
D: Identify any spatial–temporal patterns in the data by considering the spatial patterns in the weather data and the temporal patterns in the data itself.
E: Identify any spatial–temporal anomalies in the data by considering the spatial patterns in the weather data and the temporal patterns in the data itself.
Please provide the answer for each question.

---

**Issue:** Instead of performing any computation, the model merely repeats the query as a list of instructions. The response is vague, repetitive, and incomplete, lacking concrete anomaly detection, temporal reasoning, or spatial analysis.

---

**Example 3: Irrelevant and High-level steps (LLaMA-2-7B)**

**Query:** Perform a seasonality and neighbourhood analysis using the historical air quality data for the past 30 days at location yongledian_aq in the Beijing region, focusing on daily patterns. The data is recorded at 60-minute intervals. The current time is 2017-08-15 09:00:00.

**Generated Answer:**

1. Calculate the seasonality of the air quality data at location yongledian_aq using a seasonal decomposition method (e.g. STL decomposition) ...
B: Perform a time series forecasting of the air quality data at location yongledian_aq using a time series forecasting model (e.g. ARIMA, LSTM, etc.) ...
C: Investigate the impact of weather factors (e.g. temperature, humidity, wind speed, etc.) on the air quality data at location yongledian_aq using a linear regression model ...
D: Perform a clustering analysis of the air quality data at location yongledian_aq using a clustering algorithm (e.g. K–means, hierarchical clustering, etc.) ...

---

**Issue:** Instead of executing the requested analyses, the model outputs generic procedural steps without producing any computed results. Several suggested steps (e.g., time-series forecasting, clustering) are irrelevant to the original query and introduce inconsistencies across responses.

---

**Example 4: Generic, High-Level Explanations without Computation (GPT-3.5 / GPT-4)**

**Query:** What will be the traffic speed at location 409524 in the BAY region for the next 35 minutes, based on historical data from the past 1 hour recorded at 5-minute intervals? The current time is 2017-02-14 03:00:00. Please ensure that the predicted traffic speed does not exceed 100 km/h. Also, analyze how daily timestamps and neighbouring sensors impact the accuracy of traffic speed predictions.

**Generated Answer (GPT-3.5 Turbo):**

Given the historical data provided: - At 2017-02-14 03:00:00, the traffic speed at location 409524 is 64.4 km/h. - The traffic speed at location 409524 for the past hour has fluctuated between 64.1 km/h and 64.6 km/h. To predict the traffic speed at location 409524 for the next 35 minutes, we can analyze the trend in the historical data. Since the traffic speed has been relatively stable around 64.4 km/h, we can forecast that it will remain around that speed for the next 35 minutes. Therefore, the predicted traffic speed at location 409524 will be approximately 64.4 km/h for the next 35 minutes, as long as there are no major disruptions or changes in traffic patterns . . .
- Daily timestamps: Daily timestamps can impact the accuracy of traffic speed predictions by capturing the daily patterns and variations in traffic flow. For example, rush hours, weekends, holidays, and special events can all influence traffic speed . . .
- Neighboring sensors: Neighboring sensors provide additional data points that can help improve the accuracy of traffic speed predictions. By analyzing the data from neighboring sensors, we can identify correlations and patterns in traffic flow between different locations.
. . .

**Generated Answer (GPT-4):**

As an AI, I can't dynamically predict data, but I can explain how the process would work using the given data.
To predict the traffic speed at location 409524 in the BAY region for the next 35 minutes, we can use a spatiotemporal forecasting model . . .
The first step would be to analyze the historical data for sensor 409524. We can observe that the traffic speed at this sensor is relatively stable, oscillating around 64.3 km/h. This suggests that unless there are significant changes in the surrounding environment, the speed will likely remain stable for the next 35 minutes.
Next, hlwe'd analyze the neighbouring sensors data. It's important to consider neighbouring sensors because traffic conditions in one location can be influenced by conditions in nearby locations.. . .
Therefore, we can make an educated guess that the traffic speed at sensor 409524 will likely stay around its current value of 64.4 km/h for the next 35 minutes, given that there is no sudden change in traffic conditions. . . .
However, this prediction should be taken as a rough estimate. Real-time traffic conditions can be influenced by many unpredictable factors . . .

---

**Issue:** Both GPT-3.5 and GPT-4 provide qualitative explanations and high-level methods rather than executing multi-step numeric forecasts. GPT-3.5 produces a static repetition of the last observed value and supplements it with generic knowledge (e.g., "daily timestamps impact accuracy"), but does not validate constraints or generate fresh predictions. GPT-4 explicitly states its inability to perform forecasting, then offers a rough guess framed as a methodological outline. In both cases, the responses lack actual computation, constraint enforcement, and actionable reasoning.

**Example 5: Structured but Trivial Forecasts with Limited Analysis (GPT-4o Mini / DeepSeek-V3)**

**Query:** What will be the traffic speed at location 409524 in the BAY region for the next 35 minutes, based on historical data from the past 1 hour recorded at 5-minute intervals? The current time is 2017-02-14 03:00:00. Please ensure that the predicted traffic speed does not exceed 100 km/h. Also, analyze how daily timestamps and neighbouring sensors impact the accuracy of traffic speed predictions.

**Generated Answer (GPT-4o Mini):**

To forecast the traffic speed at location 409524 for the next 35 minutes based on the historical data provided and considering the neighboring sensors, we can follow these steps:
Step 1: Analyze Historical Data The historical data for the last hour . . .
Step 2: Identify Trends From the historical data, we can observe that the traffic speed has not varied significantly over the previous 55 minutes . . .
Step 3: Consider Neighboring Data The neighboring sensors (409525, 409526, 409528, 409529) also show consistent speeds, generally ranging from 61.5 km/h to 69.0 km/h . . .
Step 4: Forecasting Future Speeds Given that the speed is stable and the surrounding data from neighboring sensors does not indicate any sudden changes, we can reasonably predict that the speed at sensor 409524 will remain around the same mark.
Step 5: Create a Prediction for the Next 35 Minutes We can make a simple prediction for the next 7 readings (5 minutes each) based on the current speed and the historical average. . . .
Impact of Daily Timestamps and Neighboring Sensors: - **Daily Timestamps**: Traffic patterns can vary significantly based on the time of day . . . - **Neighboring Sensors**: Data from neighboring sensors can greatly enhance the accuracy of predictions. If speeds in neighboring areas are significantly different, it may indicate incidents, road work, or other factors affecting traffic. . . .

**Generated Answer (DeepSeek-V3):**

1. Data Analysis  Pattern Recognition
- **Target Sensor (409524) Analysis:** The traffic speed at location 409524 over the past hour has been exceptionally stable, hovering between **64.1 km/h and 64.6 km/h** . . . - **Temporal Context:** The timestamp (2017-02-14 03:00:00) is crucial. This is **3 AM on a Tuesday morning**. Traffic patterns at this time are highly predictable and characterized by the lowest volume of the day. There is no influence from morning commutes, and the road network is operating at free-flow speeds. . . .
2. Traffic Speed Prediction for the Next 35 Minutes Given the remarkable stability of the data and the time of day, the most reasonable forecasting approach is **persistence forecasting**, which assumes that the current conditions will remain unchanged in the near future. . . .
3. Impact Analysis on Prediction Accuracy a) Impact of Daily Timestamps:** During the early morning hours, traffic demand is minimal and consistent. . . . b) Impact of Neighbouring Sensors:** Traffic is a spatial phenomenon; congestion forms in waves. If sensor 409525 (upstream) shows a sudden drop in speed, it is a strong leading indicator that the congestion will likely reach sensor 409524 minutes later. . . .
Conclusion:Based on the historical stability of the data and the contextual factors of very early morning traffic, the traffic speed at location **409524** for the next 35 minutes is predicted to remain constant at **64.4 km/h**. . . .

**Issue:** While both responses are clearly structured, they rely on trivial forecasts or basic forecasting models (e.g: Persistence, Historical average or simple models like ARIMA) rather than executing advanced model-based computation. GPT-4o Mini supplements its steps with generic facts similar to other GPT variants but provides only a simple average-based prediction. DeepSeek-V3 applies contextual reasoning (time of day, spatial correlations) but still defaults to persistence forecasting, offering no multi-step modeling or constraint validation. In both cases, explanations remain descriptive rather than analytical, lacking the depth and precision required for spatio-temporal decision support.

Unlike the above baseline models, which either hallucinate patterns, repeats the query, or rely on trivial or basic model forecasts, STReason delivers precise numeric predictions using advanced

predictive models, validated against constraints and enriched with temporal and spatial attribution as shown in Example 6 below. This highlights its strength in producing interpretable, statistically grounded, and actionable outputs for spatio-temporal decision support .

---

**Example 6: Comprehensive Forecast with Temporal and Spatial Reasoning (STReason)**

**Query:** What will be the traffic speed at location 409524 in the BAY region for the next 35 minutes, based on historical data from the past 1 hour recorded at 5-minute intervals? The current time is 2017-02-14 03:00:00. Please ensure that the predicted traffic speed does not exceed 100 km/h. Also, analyze how daily timestamps and neighbouring sensors impact the accuracy of traffic speed predictions

**Generated Answer (STReason):**
Predictions: The predicted traffic speed values for the next 55 minutes at location 409524 are [64.51839, 64.49478, 64.48434, 64.4911, 64.50357, 64.51794, 64.53189]. These predictions were generated using the deep learning based GraphWaveNet Model.

Constraint Adherence: All predictions are within the traffic speed threshold, indicating that the forecasted values are within the acceptable range for traffic speed at location 409524.

Temporal Features: The Timestamp Sensitivities reveal that certain time periods have a significant impact on the traffic speed predictions. For example, timestamp 2017-02-14 02:55:00 has the highest impact, followed by timestamps representing morning rush hour such as 2017-02-14 03:00:00 and timestamps indicating evening rush hour like 2017-02-14 02:45:00. This suggests that distinct daily traffic patterns, such as rush hours, greatly influence the forecasted traffic speed values.

Spatial Features: The analysis also highlights the influence of data from neighboring nodes and significant nodes on the traffic speed predictions at location 409524. Among the significant nodes impacting the forecast, locations 401816, 401817, 409525, 404759, 404453, 409528, 404461, 409529, 409526, and 400995 play a key role in shaping the predictions. The presence of neighboring nodes like 409525, 409526, 409528, and 409529 further emphasizes the spatial factors affecting the forecast at location 409524.

In conclusion, the traffic speed predictions at location 409524 are not only influenced by temporal factors like daily traffic patterns but also by spatial factors involving data from neighboring and significant nodes. By incorporating these insights into the forecasting model, more accurate and reliable predictions can be achieved.

---

## A.11 GENERALIZABILITY OF STREASON TO OTHER SPATIO-TEMPORAL DOMAINS

While our main experiments focus on traffic and air quality domains, STReason is designed to generalize across spatio-temporal domains without modifying its core architecture or prompting strategy. To illustrate this, we tested the framework on two unseen domains: epidemiology and finance. The only adjustments involved were updating the data retrieval function to point to the relevant sources, and replacing the forecasting module with TimesNet (Wu et al., 2022) for pure time-series inputs.

---

**Epidemiology Example**

**Query:** Perform a seasonality analysis of historical national ILI case counts for the past 52 weeks. The current time is 2019-01-22 00:00:00. The data is recorded at every week. Analyze the Monthly Seasonality Strength.

**Generated Answer:** The data represents the number of ILI (Influenza-Like Illness) case counts recorded every week. The data starts with 23169 cases in the first week and peaks at 70056 cases in the 15th week before gradually decreasing to 27419 cases in the last week. The monthly seasonality strength is calculated to be 0.00036, indicating a very weak monthly seasonality pattern in the data. This suggests that the fluctuations in the number of ILI case counts do not follow a consistent monthly pattern and are more likely influenced by other factors. The low monthly seasonality strength suggests that the variations in the number of ILI case counts are not significantly affected by the month in which the data was collected. This could be due to factors such as changes in weather, vaccination campaigns, or other external factors that impact the spread of influenza-like illnesses.

---

**Finance Example**

**Query:** The current time is 2010-07-06 12:00:00. Predict the exchange rate of GBP to USD for the next 5 days using the historical data of the past 96 days with data points recorded at daily intervals. Ensure predictions stay below the 1.55 financial stability threshold. Analyze how temporal patterns affect the forecast.

**Generated Answer:** The predicted exchange rate values for the next 5 days are as follows:

$$[1.3212, \ 1.3242, \ 1.3186, \ 1.3189, \ 1.3200].$$

These predictions were generated using the deep learning-based TimesNet Model. All predictions fall within the exchange rate threshold, indicating that the model's forecasts are within acceptable limits and align with historical data. The timestamp sensitivity scores provide insights into the temporal factors influencing the exchange rate predictions. Among the timestamps analyzed, the strongest influences were observed on July 4, 2010, July 5, 2010, and July 6, 2010, with impacts of 0.19, 0.17, and 0.17 respectively. These dates likely correspond to significant events or announcements that had a notable impact on the exchange rate forecast. By analyzing these timestamps in conjunction with macroeconomic trends, we can gain a deeper understanding of the drivers behind the predicted exchange rate values.

---

These examples highlight STReason's adaptability to new spatio-temporal domains without additional training or architectural changes. Its modular design allows flexible integration of domain-specific data and models, supporting broader applicability. We plan to further extend this adaptability to multimodal settings in future work (see Section 5).

## A.12 HUMAN EVALUATION DETAILS

**Evaluation Procedure** To assess the interpretability and reasoning quality of model outputs and further validate our quantitative metrics we designed a human evaluation study involving 27 participants.

1. **Participant Selection:** Evaluators were selected based on the following criteria to ensure technical competence:

   - Minimum of a Bachelor's degree in Computing, Data Science, Statistics, Mathematics, or a related technical discipline.
   - General understanding of spatio-temporal tasks such as analysis, anomaly detection, and forecasting.

2. **Material Preparation:** The study included 18 queries covering three task categories: Analysis, Anomaly Detection, and Prediction and Reasoning. Each query was paired with two answers, one from STReason and one from a randomly selected baseline ensuring each baseline appeared an equal number of times. The order of answers and pairings was randomized to mitigate bias.

3. **Evaluation Design:** Participants were instructed to select the more effective answer based on clarity, completeness, reasoning, and overall helpfulness. They were also encouraged to provide open-ended feedback explaining their choices.

**Evaluator Background** Human evaluation was conducted involving 27 evaluators with domain relevant expertise, nearly half holding Ph.D.s and the majority specializing in Computing, Statistics, or Data Science (Figure 18).

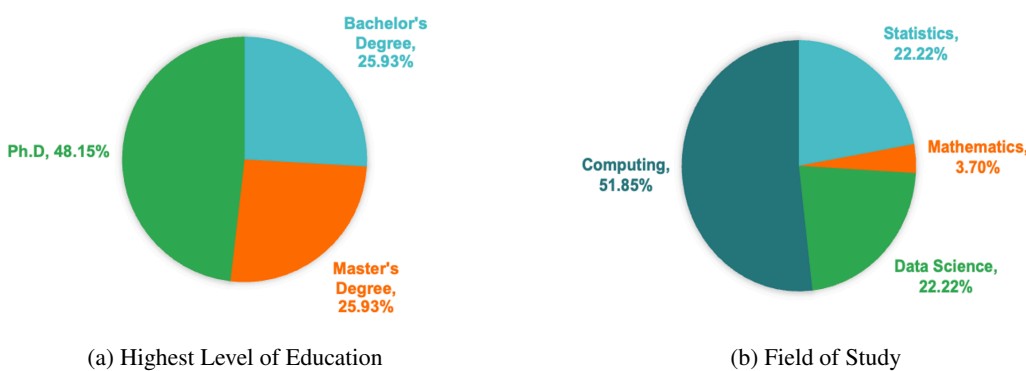

(a) Highest Level of Education                    (b) Field of Study

Figure 18: Evaluator Background

**Instructions provided to Evaluators**

Thank you for agreeing to participate as an evaluator in this critical assessment exercise. Your expertise is crucial in evaluating the performance of sophisticated question answering and inference models tailored to spatio-temporal tasks.

**Objective of the Evaluation:** The primary objective of this evaluation is to objectively assess and validate the performance of various question answering models in managing spatio-temporal inference and reasoning tasks. Your evaluations will play a pivotal role in identifying the most effective models and will guide the development of future enhancements.

**Task Overview:** You will be presented with 18 questions, each accompanied by two answers. These answers are generated by two distinct models. Please note that these answers are generated by different language models, each with unique capabilities and limitations. This might result in some answers appearing less comprehensive, less meaningful or partially incomplete.The pairings and the order of the answers have been randomized to ensure an unbiased assessment.

**Your Role:**

- **Answer Selection:** For each question, your task is to select the answer that most comprehensively and accurately addresses the query. There is no right or wrong choice, only your professional judgment on which answer performs better in the context of the given question. While making your selections, you can consider the following aspects:
    - Adherence to Constraints: Evaluate how well the answer adheres to the specific constraints set by the question.
    - Completeness and Accuracy: Assess whether the answer fully captures all necessary aspects of the query and provides accurate information.
    - Logical Progression and Clarity: Determine the logical flow and clarity of the explanation and reasoning within the answer.
- **Providing Feedback:** After evaluating each answer, please provide qualitative feedback in the space provided explaining why you chose one answer over the other. This feedback is invaluable as it helps us understand the reasoning and decision-making processes that influenced your preferences.

**Confidentiality and Ethical Consideration:** Please treat the information provided during this evaluation with confidentiality. The data, questions, and responses should not be discussed outside of this evaluation context.

