# OpenReview forum: "A Modular Multi-task Reasoning Framework Integrating Spatio-temporal Models and LLMs"
_ICLR.cc/2026/Conference — Submitted to ICLR 2026_

### Official Review · Reviewer_69yp · 2025-10-31

**Soundness:** 3
**Presentation:** 3
**Contribution:** 2
**Rating:** 4
**Confidence:** 2

**Summary:**

This paper focuses on long-contextual question answering, which requires spatio-temporal cross-task reasoning.

This paper proposes STReason, which utilizes LLM as a high-level planner and incorporates spatio-temporal models as tools/functions to deliver subtask decomposition and tool calling to handle multi-task, long-form inference and execution.

The authors introduce a new benchmark specifically for spatio-temporal tasks; both automated and human evaluations demonstrate that STReason has significant advantages over robust LLM baseline algorithms.

**Strengths:**

The paper proposes a new framework for spatio-temporal reasoning. Combining LLMs’ general knowledge with domain-specific small models is effective: the small models complement LLMs with specialized expertise while LLMs provide a unified framework for high-level scheduling, integrating, and analyzing information gathered by small models.

The experimental results show their superior performance, especially in the metric of Factuality score and human evaluation.

**Weaknesses:**

1. It would significantly strengthen the evaluation to include domain-specific baselines in spatio-temporal reasoning. Such comparisons would better contextualize the gains and clarify where the proposed approach provides unique value.

2. It’s unclear whether the function pool actually helps. Intuitively, it should, but the results in Table 3 don’t clearly demonstrate a benefit. Additionally, how to choose domain-specific functions or models for LLMs.

3. While the framework is tailored to spatio-temporal settings and has some novel elements, the overall paradigm—LLMs as tool callers invoking domain-specific small models—feels familiar. As a result, the contribution may fall a bit short of the typical acceptance bar.

4. Additionally, it is not clear how the proposed benchmark can contribute to the research field. The author states that they address long-form reasoning, but I did not see a lot of description or comparison about this.

**Questions:**

It appears that answers to the reasoning tasks are free-form. Could the authors provide more details about the evaluation protocol? If an additional LLM is used as the evaluator, how is the consistency and accuracy of its judgments ensured?

---

> ### Author Response · Authors · 2025-11-18
> **Addressing Weaknesses : W1**
>
> ### **Summary of Response**
> We thank the reviewer for highlighting the strengths of STReason, especially the effectiveness of combining LLMs with specialized spatiotemporal models, and the strong improvements in Factuality and human evaluation. Below, we address concerns regarding domain-specific baselines, the benefit of the Function Pool and selection of its corresponding modules, novelty and significance of the overall paradigm, contributions of the benchmark, and evaluation protocol for free-form reasoning outputs.
>
> ### **W1: Need for domain Specific Baselines**
> We clarify below why domain-specific spatiotemporal models cannot serve as meaningful baselines for our problem setting:
>
> **Existing Domain-specific Spatiotemporal Models are Predictors, not Reasoning Models**: Existing domain specific spatiotemporal models such as STGCN, GraphWaveNet, AirFormer, AirPhyNet etc. are designed exclusively for numeric prediction. They:
> - take structured tensors as input,
> - output only vectors/scalars,
> - cannot interpret natural-language queries,
> - cannot perform multi-step analysis (trend, seasonality, anomaly reasoning, constraint enforcement), and
> - cannot generate long-form explanations or reasoning chains.
>
> While such models can serve as analytical backends within STReason (like we use GraphWaveNet inside the forecast() module), they cannot serve as meaningful baselines for evaluating language-based, multi-task, long-form QA setting.
>
> In addition to domain specific models we also explored baselines across two other categories to enable comprehensive comparison with STReason:
> - **Program-based Models (e.g., VISPROG, ViperGPT, PAL)**: As explained in Appendix A.1, there are multiple program-based hybrid approaches introduced for different tasks. As shown in Table 4, these cannot be compared because:
>     - they operate on different modalities (images, code, tables),
>     - they lack numerical analytics (trend, seasonality, anomaly detection),
>     - they output short categorical decisions (object labels or binary decisions) rather than structured reasoning, and
>     - they cannot execute STReason style queries due to missing spatiotemporal primitives.
>
>     When we tested representative STReason queries on several open-source hybrid models, none could execute or return valid answers due to mismatched modalities and missing functional primitives (Appendix A.1). This makes direct  comparison infeasible.
> - **Spatiotemporal aware LLMs (e.g., UrbanGPT, UrbanLLM, UniST, ST-LLM+)**: Recent works integrate LLMs into spatiotemporal workflows, but primarily to improve forecasting accuracy, not to perform reasoned analytical responses. They **do not**:
>     - handle multi-task analytical workflows,
>     - produce coherent long-form explanations, or
>     - support constraint-driven reasoning.
>
>     The closest prior work, UrbanLLM, addresses urban QA but generates short, surface-level responses without  coherent long-form reasoning. Hence these models are also not directly comparable.
>
> **Why General-Purpose LLMs are the appropriate Baselines**  Due to the absence of existing systems capable of spatiotemporal multi-task execution and generating long-form answers, we compared STReason against strong general-purpose LLMs, which represent the most capable and scalable reasoning models currently available. These models:
> - accept the same natural-language input,
> - attempt to answer the full free-form reasoning queries, and
> - allow us to isolate the contribution of STReason's program-guided architecture.
>
> We view **STReason as a foundational system for this emerging direction of spatiotemporal reasoning**, analogous to how models like  VISPROG and PAL advanced visual and mathematical reasoning. By extending the program-based reasoning paradigm to spatiotemporal analytics, we aim to **establish a foundation for future tool-augmented or agent-style models**. We will publicly release our benchmark and evaluation code to facilitate direct comparison with forthcoming models that integrate structured reasoning and domain-specific tools.

---

> ### Author Response · Authors · 2025-11-18
> **Addressing Weaknesses : W2**
>
> ### **W2: Significance of the Function Pool and Definition of its' corresponding Modules**
> We thank the reviewer for raising this point. Below, we clarify the significance of the Function Pool and how it grounds STReason’s program generation in explicit, well-defined semantics.
>
> **Significance of the Function Pool:** As described in Appendix A.2, the Function Pool is the explicit grounding mechanism that enables STReason to perform reliable program synthesis. It is intentionally designed as a declarative, modular registry that maps each function to its natural language description, argument schema, and output format providing the LLM with:
> - semantic grounding (what each operation does) and
> - structural constraints (how arguments must be composed)
>
> This design keeps the Function Pool lightweight, compositional, and easily extensible.
>
> Empirically, we demonstrate the importance of the Function Pool in two complementary ways:
>
> 1. **Ablation Study of Function Pool (Table 3)**: This experiment highlights the benefit of augmenting in-context examples with a curated function pool across all example construction strategies. As the reviewer noted, the gains are smaller in Equal, Random and Test Query Include settings because the in-context examples include the same task type as the test query. This naturally reduces the marginal benefit of the Function Pool, reflecting complementarity rather than redundancy.
>
>     However, the benefit becomes pronounced in the Test-Query-Exclude setting, where we remove all examples sharing the same task type as the test query. Even under this strict condition, STReason continued to generate correct and executable programs with strong accuracy with the presence of the function pool. This clearly demonstrates that when task-specific cues are absent, the Function Pool serves as the only source of grounded functional semantics, enabling the LLM to infer valid reasoning steps.
>
> 2. **Cross-Domain Generalization (Appendix A.11)**: We also tested STReason on two unseen domains, finance and epidemiology, without modifying either the architecture, function pool structure or the existing in-context examples. STReason successfully generated structured and executable programs that adapted arguments and semantics to each new domain. This demonstrates that:
>     - Generalization is driven by functional grounding, not example memorization
>     - The Function Pool enables compositional, domain-agnostic reasoning
>
> This result further confirms that the Function Pool is a core mechanism enabling STReason to handle new tasks and domains in a zero-shot manner.
>
> **How Functions/Models Are Selected:** Functions in the pool correspond to fundamental analytical sub-tasks in spatiotemporal workflows (e.g., trend analysis, seasonality, anomaly detection, forecasting, etc,). Each module wraps a well-established statistical method or a SOTA pretrained model, ensuring reliability without requiring task-specific training for each query. For example:
> - *forecast()* module embeds a pretrained GraphWaveNet model
> - *analyze_trend()* module applies seasonal decomposition and linear regression
> - *detect_anomaly()* leverages unsupervised z-score based techniques
> - *gen_explanation()* uses an LLM guided approach to synthesize intermediate results into natural language.
>
> All module definitions are documented in Appendix A.4, with implementations openly available in step_funcs.py in our shared repository.
>
> The selected modules satisfy are widely used in real-world spatiotemporal analytics and represent reusable operations across multiple domains. Importantly, new domain-specific functions can be added or existing modules can be replaced or removed through simple declarative entries without modifying the LLM or retraining.

---

> ### Author Response · Authors · 2025-11-18
> **Addressing Weaknesses : W3**
>
> ### **W3: Novelty and Scientific Significance of STReason**
> We thank the reviewer for this point. While STReason shares the high-level idea of “LLM + tool calling,” we respectfully argue that it introduces several domain-specific and scientifically meaningful advances that go beyond existing paradigms.
>
> **Scientific Significance of STReason:** STReason introduces four contributions not addressed in prior literature:
> 1. **Long-Form, Multi-Task Answer Generation:** Unlike models narrowly focused on numeric forecasting or short/MCQ QnA, STReason generates coherent, long-form explanations that combine numerical analysis, reasoning chains, constraint checks, and contextual interpretation (Lines 70–73). This addresses a major gap in spatiotemporal AI, where outputs are typically brief and non-explanatory.
> 2. **Pioneering Program-based Reasoning Architecture for Spatio-temporal Tasks:** As discussed in Section 2, prior program-based systems target mainly vision and tabular domains. STReason is the first to extend this paradigm to the spatio-temporal domain that introduces unique challenges such as handling heterogeneous inputs (e.g., time-indexed arrays, geospatial variables), applying domain constraints (e.g., speed thresholds), and incorporating auxiliary metadata. Unlike existing systems that analyze static images or tables, STReason tackles the challenge of aligning spatio-temporal data to resolve complex analytical queries.
> 3. **Systematic Evaluation Framework for Long-form Spatio-temporal Reasoning:** In Section 4.1, we introduce three new metrics to evaluate long-form answers, grounded in the principles of QnA literature, specifically tailored to spatio-temporal reasoning outputs. These metrics go beyond traditional forecasting metrics like RMSE or MAE by capturing the logical, factual, and structural quality of model outputs. This contribution offers a timely and meaningful addition to the growing landscape of question answering research.
> 4. **Generalization without Fine-Tuning:** STReason shows how pretrained ST models and LLMs can be effectively integrated and applied to structured reasoning tasks without task/domain-specific fine-tuning. Adding new tasks requires only updating a Function Pool entry, not retraining which yields a practical, modular blueprint for future reasoning systems.
>
> Together, these contributions establish STReason as a methodological advancement in structured, interpretable spatiotemporal reasoning.
>
> **Novelty Relative to Existing “LLM + Tool Calling” Systems:** While inspired by modular reasoning, STReason differs from prior work in several fundamental ways:
> - **Different domain & problem class:** Existing systems (VISPROG, ViperGPT, Chain-of-Table, PAL, HuggingGPT) target vision, tabular QA, or symbolic/mathematical problems. STReason is a pioneering work to extend the modular reasoning paradigm to the structured spatio-temporal domain; a setting that introduces entirely different challenges. Unlike existing systems that analyze static images or tables, STReason tackles the challenge of aligning spatio-temporal data to resolve complex analytical tasks generating interpretable, well-reasoned outputs. These capabilities lie beyond the scope of existing LLM-based tool calling frameworks as highlighted in Section 1 and Section 2.
> - **Execution on real-world structured data:** STReason executes real-world queries over structured spatio-temporal data such as traffic flow and air quality requiring distinct modeling techniques. In contrast, other models operate on fundamentally different inputs modalities.
> - **Execution on Diverse Query Types:** STReason handles analytical queries which require structured data processing, spatio-temporal modeling, and contextual reasoning while other LLM-based tool calling models handle different types of straight forward queries. Example:
>     - VISPROG: *“How many muffins can each kid have in the picture?”*
>     - STReason: *“Perform a trend, seasonality, and neighborhood analysis of the historical traffic speed data for the past 90 days at location 402117 in the BAY region.”*
> - **Analytical and Explanatory Outputs:** Unlike other methods that produce short categorical responses (e.g., object labels, counts, or binary decisions), STReason generates long-form, interpretable outputs combining structured numerical analysis, constraint evaluation, and explanatory narratives. For a concrete illustration, we kindly refer you to the demo video linked in Line 122 of the paper.
>
> In summary, while we build upon the modular QnA paradigm, STReason advances this framework into a novel domain with distinct challenges and tasks, introducing mechanisms for structured, actionable reasoning in urban decision-making. We therefore believe **STReason represents a meaningful and novel contribution, analogous to how models like VISPROG and PAL advanced visual and mathematical reasoning.**

---

> ### Author Response · Authors · 2025-11-18
> **Addressing Weaknesses : W4**
>
> ### **W4: Clarifying the Contribution and Importance of Our Benchmark Dataset**
> We thank the reviewer for raising this point. We clarify below how our benchmark contributes to the field and how it enables the evaluation of long-form spatiotemporal reasoning, which existing datasets cannot support.
>
> **Limitations of Existing Spatiotemporal Datasets:** Existing spatiotemporal datasets such as those used for traffic forecasting (e.g: METR-LA, PEMS-BAY) or air quality prediction (e.g: Beijing Air, KnowAir) are designed exclusively for numerical tasks such as prediction. They:
> - contain no natural-language questions
> - require no explanation or reasoning steps
> - evaluate only single-task prediction using RMSE/MAE
> - provide no ground truth for explanations or constraint handling
> - include no mechanism to evaluate factual consistency, interpretability, or reasoning quality
>
> As a result, to our knowledge no existing dataset in the literature can evaluate long-form reasoning over spatiotemporal data. This is the precise gap our benchmark fills.
>
> **Significance of our Benchmark to the Research Community:** Our benchmark is specifically designed to assess long-form, multi-task spatio-temporal reasoning. It introduces several components missing from prior work:
> - **Multi-task natural language queries** that cover three distinct categories: analysis, forecasting with reasoning, and anomaly detection with reasoning. This shifts evaluation from mere “predictions” to structured reasoning over spatiotemporal data.
> - **Grounded structured programs** that reflect the step-by-step reasoning process. This enables reproducible, tool-augmented reasoning evaluation.
> - **Ground-truth answers**, annotated with the key analytical and reasoning components necessary for producing interpretable, comprehensive outputs.
>
> This structure enables fine-grained evaluation of multi-step reasoning quality, not just numeric accuracy and establishes a standardized testbed for future spatiotemporal reasoning research. Upon acceptance, we will publicly release the full benchmark dataset. As part of future work, we also plan to expand it to additional domains such as weather, finance, and epidemiology, further broadening its applicability and value to the spatiotemporal reasoning research community.
>
> **Evaluation of Long-form Reasoning:** As explicitly stated in the manuscript, STReason produces long-form responses. Due to the lack of standardized metrics exist for systematically evaluating long-form question answering, we introduced three metrics designed specifically for this setting (Section 4.1). These metrics collectively evaluate reasoning structure, factual grounding, and interpretability, going far beyond prediction accuracy.
>
> To further validate the performance of STReason beyond automated metrics, we also conducted a rigorous human evaluation to assess the credibility, clarity, and reasoning quality of model generated answers. This complements our quantitative results with human judgment, thereby offering deeper insights into the effectiveness of STReason in real world contexts. STReason consistently outperformed baselines, validating the practical value of long-form reasoning.

---

> ### Author Response · Authors · 2025-11-18
> **Response to Question : Q1**
>
> ### **Q1: Evaluation Framework & Use of LLM as Evaluator**
> We thank the reviewer for this important question. Below we clarify our evaluation framework for free-form reasoning outputs and how we ensure the consistency and reliability of LLM-based judgments.
>
> **Evaluation Protocol for Free-Form Long-Form Answers:** As described in Section 4.1 and Appendix A.7, we evaluate model-generated explanations using three structured metrics, each designed specifically for long-form spatiotemporal reasoning:
> - **Constraint Adherence**: A deterministic True/False check verifying whether all query-specified constraints are satisfied (e.g., weekend-only filtering, speed thresholds, time-window correctness). We use a fixed-format verifier prompt (Appendix A.7, Fig. 15).
> - **Factuality Score:** The evaluator extracts required analytical components from the query, compares them to the ground truth, and computes (No: of correct components) / (No: of expected components). This yields a precise, grounded measure of reasoning correctness (Appendix A.7, Fig. 16).
> - **Coherence Score:** A rubric-based score assessing logical flow, clarity and cohesiveness of the explanation (Appendix A.7, Fig. 17).
>
> These three metrics collectively assess completeness, factual correctness, and reasoning quality, while remaining robust to variations in wording and style which cannot be measured by BLEU/ROUGE-style metrics.
>
> **Ensuring Consistency & Accuracy When Using an LLM as Evaluator:** We acknowledge the reviewer's concern about using an LLM as an evaluator and clarify that multiple safeguards were put in place to ensure reliability:
> - **Highly Structured & Constrained Evaluator Prompts:** All evaluator prompts follow strict, deterministic templates (Appendix A.7, Figures 15–17). The prompts force the evaluator to read the ground truth, extract key components, check each point explicitly, return an unambiguous output format. This significantly reduces subjectivity and prevents drift in the evaluation.
> - **Objective Criteria:** The evaluator is never asked to “rate” in open-ended prose. Instead, it performs, binary checks, component matching and fixed-scale scoring. All scoring dimensions are discrete, well-defined, and easy to verify. The constrained output formats (True/False, component counts, 1–3 coherence score) leads to consistent scoring across runs.
> - **Ground-truth reference anchoring:** The evaluator always compares model output directly against structured ground-truth components, reducing subjective drift.
> - **Empirical Alignment with Human Evaluation:** Most importantly, as shown in Section 4.3, the ranking of models by human preference mirrors the ordering produced by automated metrics. This strong correlation validates that the LLM-based judgments are consistent with human judgments. This  provides empirical evidence of the reliability of the automated evaluator.
>
>
> **Why an LLM Evaluator is appropriate in this Context:** Spatio-temporal long-form answers cannot be evaluated by conventional metrics like BLEU, ROUGE, or embedding similarity because;
> - Answers contain both narrative explanation and numerical analysis,
> - Ground-truth answers include structured components and
> - Wording may differ while reasoning quality remains equivalent.
>
> Thus, a structured LLM-based evaluator provides the only practical mechanism for systematically assessing logical correctness in long-form analytical outputs. Our design follows best practices in long-form QnA evaluation similar to recent LLM-as-judge studies in the literature.

---

> ### Author Response · Authors · 2025-11-24
>
> Dear Reviewer 69yp,
>
> We sincerely appreciate the time and effort you have dedicated to reviewing our work. We have carefully addressed your questions and concerns above and are happy to continue the discussion should you have any further questions or require clarifications.
>
> If there are no further questions, we kindly request you to consider revisiting your score in light of our responses, as we are approaching the end of the rebuttal period.
>
> Thank you once again for your valuable feedback and for helping us improve our work !

---

> ### Author Response · Authors · 2025-11-27
>
> Dear Reviewer 69yp,
>
> We would like to kindly follow up regarding our previous response to your comments.
>
> As the discussion period concludes on December 2, we would be grateful if you could consider revisiting your overall assessment  including both the score and the confidence rating in light of the responses we have provided. We sincerely appreciate your time and constructive feedback, and we remain happy to clarify any remaining concerns.
>
> Please let us know if you have any further questions or if additional clarification would be helpful from our side.
>
> Thank you once again for your valuable feedback and for helping us improve our work !

---

> > ### Comment · Reviewer_69yp · 2025-11-27
> >
> > Thanks for the authors’ efforts in addressing my concerns. I believe my original score already reflects my opinion of this work, so I will keep it unchanged.

---

> > > ### Author Response · Authors · 2025-11-28
> > >
> > > Thank you very much for taking the time to review our additional clarifications. We appreciate your thoughtful engagement with the work.
> > >
> > > To ensure our responses fully addressed your major concerns, we would like to briefly recap the key points we provided and also include some additional empirical evidence:
> > > - **Domain-specific baselines:** We clarified why existing spatiotemporal models (e.g., STGCN, GraphWaveNet, AirFormer) cannot serve as meaningful baselines for multi-task, long-form reasoning, and why general-purpose LLMs offer the most appropriate comparison for our setting.
> > > - **Function Pool effectiveness:** We provided ablations and cross-domain evidence, including results in new unseen domains, showing that the Function Pool materially improves reasoning robustness, especially when task-aligned examples are absent. Beyond these, we further constructed two new evaluation sets (20 epidemiology queries, 20 finance queries) to directly test zero-shot generalization. STReason achieved substantially higher Factuality Score compared to the strongest baseline (DeepSeek-V3), reinforcing that its program-guided architecture generalizes reliably to entirely new domains:
> > > | | **Epidemiology Data** | **Finance Data** |
> > > | ------------------ | ----------------------- | -------------------------- |
> > > |   DeepSeek-V3     |          26.28%      |   43.33%                     |
> > > | STReason            | **83.05%**           | **87.5%**                 |
> > >
> > > These results confirm that STReason generalizes well beyond the original benchmark, even without modifying prompts, architecture, or examples.
> > > - **Novelty of the paradigm:** We explained how STReason extends program-based reasoning into the spatiotemporal domain for the first time, enabling structured analytical workflows, constraint handling, and long-form explanations that are not addressed in prior “LLM + tools” systems.
> > > - **Benchmark contribution:** We clarified how the benchmark fills a real gap in existing datasets by providing multi-task natural-language queries, structured programs, and ground-truth reasoning components, necessary for evaluating long-form spatiotemporal reasoning.
> > >
> > > Please let us know if there are any aspects you feel are not yet fully addressed or would benefit from further clarification.   We would be happy to provide additional details.
> > >
> > > Thank you again for your time and constructive insights!

---

### Official Review · Reviewer_PRfp · 2025-11-01

**Soundness:** 3
**Presentation:** 3
**Contribution:** 3
**Rating:** 6
**Confidence:** 3

**Summary:**

The paper presents STReason, a two stage modular spatio temporal reasoning framework where an LLM emits executable ST Programs and an interpreter runs specialized modules to produce verifiable execution rationales. Experiments on a new benchmark show improvements over pure LLM baselines in constraint adherence, factuality, and forecasting accuracy; contributions include the architecture, an extensible function pool, and evaluation suite, with limitations in manual example/function curation and rare anomaly detection.

**Strengths:**

1. This paper proposes a new benchmark covering spatio-temporal analysis, anomaly detection, and forecasting reasoning, with unified evaluation metrics that better reflect practical task and engineering constraints.
2. It provides extensive empirical results on automatic metrics and human evaluation, and ablation studies that reveal how key design choices (e.g., in‑context examples and the Function Pool) concretely affect performance.

**Weaknesses:**

The baseline setup is somewhat simplistic; it lacks direct comparisons with stronger time‑series/spatio‑temporal specialized methods or advanced hybrid baselines that invoke external tools, which may lead to an overestimation of the method’s relative advantage.

The approach depends on a manually designed and maintained Function Pool (and in‑context examples), reducing automation and deployability; when handling tasks with strict constraints, repeated Function calls can cause loss of contextual information. From the second demo in the video, when asked to query weekend data, the RefineOutput function appears to forget this requirement.

**Questions:**

See the weakness above.

---

> ### Author Response · Authors · 2025-11-17
> **Addressing Weaknesses : W1**
>
> ### **Summary of Response**
> We thank the reviewer for recognizing the novelty of the unified benchmark, the comprehensiveness of the empirical evaluation, and the clarity of STReason’s design. In response to the reviewer’s concerns, we clarify the rationale behind the baseline setup, design and extensibility of the Function Pool and in-context examples, and the issue raised regarding contextual consistency in the video.
>
> ### **W1: Rationale for Baseline Selection**
> We appreciate the reviewer’s thoughtful comment regarding the choice of baselines and below we clarify our rationale.
>
> We systematically evaluated three categories of potentially comparable methods:
> - **Hybrid “LLM + external tool” Models (e.g., VISPROG, ViperGPT, PAL)**: As explained in Appendix A.1, there are multiple program-based hybrid approaches introduced for different tasks. However, these systems differ fundamentally from our setting as they operate over visual or symbolic inputs, produce object labels or binary decisions, and lack numerical analytics or long-form explanation capabilities. Table 4 shows that direct comparison is not feasible due to mismatched modalities, query types, and output formats. Furthermore, when we tested representative STReason queries on several open-source hybrid models, none could execute or return valid answers due to mismatched modalities and missing functional primitives (Appendix A.1). This makes direct  comparison infeasible.
> - **Domain-specific Spatiotemporal Models (e.g., STGCN, GraphWaveNet, AirFormer, AirPhyNet)**: These models are designed exclusively for numeric prediction and output only scalars or vectors. They cannot parse natural-language queries, perform multi-step reasoning, or generate interpretable explanations. While such models can serve as analytical backends within STReason (like we use GraphWaveNet inside the forecast() module), they cannot serve as meaningful baselines for evaluating language-based, multi-task, long-form QA setting.
> - **Spatiotemporal aware LLMs (e.g., UrbanGPT, UrbanLLM, UniST, ST-LLM+)**: Recent works integrate LLMs into spatiotemporal workflows, but primarily to improve forecasting accuracy, not to perform reasoned analytical responses. Although these models incorporate LLMs, they remain focused primarily on numeric forecasting and do not support structured explanation or reasoning. The closest prior work, UrbanLLM, addresses urban QA but generates short, surface-level responses without  coherent long-form reasoning. Hence these models are also not directly comparable in functionality.
>
> **Why general-purpose LLMs are the appropriate baselines:** Accordingly, due to the absence of existing systems capable of spatiotemporal multi-task execution and generating long-form answers, we compared STReason against strong general-purpose LLMs, which represent the most capable and scalable reasoning models currently available. These models accept the same natural-language inputs as STReason and ensures that any performance gains can be directly attributed to STReason’s program-guided architecture. Across all baselines, STReason delivers consistent gains in factuality, coherence, and constraint adherence, demonstrating the improvements arise from our program-guided modular design, not simply “LLMs with tools.”
>
> That said, we view **STReason as a foundational system for this emerging direction of spatiotemporal reasoning**, analogous to how models like  VISPROG and PAL advanced visual and mathematical reasoning. By extending the program-based reasoning paradigm to spatiotemporal analytics, we aim to establish a foundation for future tool-augmented or agent-style models. We will publicly release our benchmark and evaluation code to facilitate direct comparison with forthcoming models that integrate structured reasoning and domain-specific tools.

---

> ### Author Response · Authors · 2025-11-17
> **Addressing Weaknesses : W2**
>
> ### **W2: Manual Function Pool + In-Context Examples and Issue Highlighted in the Demo Video**
> We thank the reviewer for raising these points. We address the concerns on the Function Pool design, the role of in-context examples, and the potential misunderstanding of the demo video below:
>
> **Existing Function Pool Design and Extensibility:** STReason’s Function Pool is intentionally designed as a declarative, modular registry that maps each function to its natural language description, argument schema, and output format (Appendix A.2). This plug-and-play design allows new analytical operations to be added without retraining or modifying the program generator, typically using only a few lines of declarative configuration. The resulting **Function Pool is lightweight, compositional, and easily extensible.**
>
> That said, we agree that manual maintenance may become a bottleneck as the system grows. This is why we have explicitly included automated Function Pool maintenance as part of our future work (Section 5). Specifically, we plan to explore:
> - **LLM-Guided Function Induction:** Use the LLM to automatically inspect interpreter libraries and infer new function definitions, allowing dynamic Function Pool expansion without human intervention.
> - **Usage-Based Pruning:** Track invocation statistics to identify and remove under-utilized functions, keeping the pool efficient and adaptive.
>
> These are high-level directions that would require further research and validation, but we believe they highlight promising avenues for improving long-term scalability and usability.
>
> Meanwhile, we also note that the current declarative format of the Function Pool supports simple, scalable updates where new functions can be added or modified with minimal engineering effort, and without changes to the core program generation or execution logic. We believe this design is a strong foundation for incorporating automated maintenance tools in the future.
>
> **Role of In-context Examples and Extensibility:** In-context examples in STReason serve as format and composition cues, illustrating how functions can be sequenced. Evidence:
> - **Test-Query-Exclude ablation (Table 3):** Here we removed all examples sharing the same task type as the test query and STReason still generated correct and executable programs with strong accuracy. This shows that the framework can infer the correct reasoning flow purely from functional semantics, without relying on task-aligned examples.
> - **Generalization to Unseen Domains (Appendix A.11):** Without modifying either the architecture or the existing in-context examples, STReason successfully generated structured and executable programs that adapted arguments and semantics in entirely new domains. This demonstrates that generalization is driven by structured function semantics, not example memorization.
>
> That said, as noted in Section 5, we plan to automate example generation for new tasks that lie beyond the three major types of tasks we've evaluated. This will further reduce manual curation and enhance scalability to additional task families.
>
> **Clarifying the Concern on Context Preservation in Demo Video:** We respectfully clarify that the concern raised is caused by a misunderstanding. In the second example of the demo video, the ST-Program generated includes the IMPOSE_CONSTRAINTS() function which filters the dataset to include weekend timestamps only, before anomaly detection is performed. The corresponding Execution Rationale confirms that the constraint was enforced stating *“Imposed constraints and retrieved data from weekends only.”*
>
> This confirms that the constraint was not dropped at any point and the model handled the contextual information promptly. The REFINE_OUTPUT() function is used to output the final result from any task in a standardized format (summary string, table, or numeric result) depending on the task context. This step simply formats the final output and it does not re-describe constraints, but the underlying execution fully respects them.
>
> Hence, there's **no loss of contextual information during repeated function calls** in the demonstrated example.

---

> ### Author Response · Authors · 2025-11-24
>
> Dear Reviewer PRfp,
>
> We sincerely appreciate the time and effort you have dedicated to reviewing our work. We have carefully addressed your questions and concerns above and are happy to continue the discussion should you have any further questions or require clarifications.
>
> If there are no further questions, we kindly request you to consider revisiting your score in light of our responses, as we are approaching the end of the rebuttal period.
>
> Thank you once again for your valuable feedback and for helping us improve our work !

---

> ### Author Response · Authors · 2025-11-28
>
> Dear Reviewer PRfp,
>
> We would like to kindly follow up regarding our previous response to your comments.
>
> As the discussion period concludes on December 2, we would be grateful if you could consider revisiting your overall assessment including both the score and the confidence rating based on the responses we have provided. We sincerely appreciate your time and constructive feedback, and we remain happy to clarify any remaining concerns.
>
> Please let us know if you have any further questions or if additional clarification would be helpful from our side.
>
> Thank you once again for your valuable feedback and for helping us improve our work !

---

### Official Review · Reviewer_2E98 · 2025-11-01

**Soundness:** 3
**Presentation:** 3
**Contribution:** 3
**Rating:** 4
**Confidence:** 2

**Summary:**

This paper introduces STReason, a new framework that fuses LLMs with specialized spatio-temporal models to perform multi-task reasoning and execution without task-specific fine-tuning. STReason operates in the command generation stage and command execution stage. In the first stage, it generates a ST-Program based on examples, and on the second stage it executes the ST-Program sequentially using 12 modular components. The authors built a new benchmark of 150 instances from real-world datasets and evaluated the STReason on the proposed benchmark, along with baselines LLMs. Results show that the proposed STReason can improve the performance on the proposed benchmark.

**Strengths:**

1. The transparency and reproducibility is great. There's an open source repo and a video demo in the paper, verifying that the method is working and also ensuring it's easy to reproduce / apply the paper.
2. The STReason proposal is novel. It combines LLM reasoning with domain-specific spatio-temporal models for multi-task inference and explanatory reasoning.
3. The system design is simple and extensible.
4. There are human annotators validating the accuracy of the proposed dataset.

**Weaknesses:**

1. The performance of the proposed approach largely depend on manually curated in-context examples. As a result, the generalization is greatly limited.
2. The evaluation dataset is small (150 queries) and spans only among traffic and air quality, which limits the evaluation on generalization.
3. The baselines used are all general-purpose LLMs, but no comparison is made with specialized spatio-temporal neural networks. The comparison with specialized spatio-temporal neural networks will help to establish a context for the performance that LLM achieved.

**Questions:**

1. How are Command Interpreter Modules determined? Why do we include one module instead of another? Are those just selected by authors or there are some rational behind the selection.

---

> ### Author Response · Authors · 2025-11-17
> **Addressing Weaknesses : W1, W2**
>
> ### **Summary of Response**
> We thank the reviewer for recognizing the novelty, transparency, extensibility, and reproducibility of STReason and appreciate the constructive feedback. Below, we address concerns regarding dependence on in-context examples, dataset scale and domain diversity, baseline choice, and the rationale behind module selection.
>
> ### **W1: Role of in-context examples and Related Concerns on Generalizability**
> We clarify that, STReason’s generalization is not solely dependent on manually curated in-context examples. These examples provide format and composition cues, while the core reasoning ability is driven by the program-guided architecture integrating the Function Pool, Command Generator, and Command Interpreter.
>
> **In-context Examples are not solely the source of reasoning capability:** Within STReason Framework, in-context examples illustrate how functions are sequenced, whereas the Function Pool and Interpreter ground each operation in explicit functional definitions. Evidence:
> - **Test-Query-Exclude ablation (Table 3):** Here, all examples sharing the same task type as the test query are removed and STReason still generated correct and executable programs with strong accuracy. This shows that the framework can infer the correct reasoning flow purely from functional semantics, without relying on task-aligned examples.
> - **Generalization to Unseen Domains (Appendix A.11):** Without modifying either the architecture or the existing in-context examples, STReason successfully generated structured and executable programs that adapted arguments and semantics in entirely new domains. This demonstrates that generalization is driven by structured function semantics, not example memorization.
>
> That said, as noted in Section 5, we plan to automate example generation for new tasks that lie beyond the three major types of tasks we've evaluated. This will further reduce manual curation and enhance scalability to additional task families.
>
> ### **W2: Dataset Scale and Domain Coverage**
> We appreciate the reviewer’s concern regarding dataset scale. As discussed in the manuscript, no standardized benchmark currently exists for evaluating multi-task, long-form, interpretable spatiotemporal reasoning. Unlike prior datasets, which typically focus on single-task numerical outputs, we constructed this benchmark dataset from scratch as the first systematic evaluation suite tailored for such reasoning tasks.
>
> The benchmark size and domain selection were deliberately chosen for methodological soundness rather than data volume, guided by the following principles:
> 1. **Designed for Breadth over quantity:** STReason is a training-free framework and does not learn parameters from data but evaluates how well a program-guided reasoning system generalizes across diverse spatiotemporal settings. Consequently, the benchmark’s purpose is not statistical representativeness but conceptual coverage, testing reasoning accuracy, constraint adherence, factuality, and interpretability across varied spatiotemporal conditions.This requires carefully curated instances rather than large sample sizes.
> 2. **Dense Reasoning Complexity in each instance:** All 150 instances includes (i) a natural-language query, (ii) a step-by-step executable program, and (iii) a verified ground truth  output. Constructing and validating these triples required extensive human annotation and cross-checking of intermediate reasoning steps, ensuring correctness and logical coherence at every stage. Each instance encodes multiple reasoning factors including spatial and temporal dependencies, forecasting horizons, and domain constraints making the dataset dense in reasoning diversity despite modest scale.
> 3. **Representative cross-domain coverage:** As described in Section 4.1, the benchmark already spans three fundamental spatiotemporal tasks across two real-world domains: Traffic (PEMS-BAY and METR-LA) and Air quality (Beijing and Shenzhen). These domains were chosen because they embody distinct spatiotemporal dynamics, forming a robust foundation for evaluating cross-domain generalization. As shown in Appendix A.11, STReason generalizes to unseen domains (finance and epidemiology) without modifying examples or architecture. This demonstrates that the benchmark is sufficient to evaluate core generalization capabilities.
>
> While our dataset covers two major domains and three of the most representative tasks in the domain of Spatiotemporal Data Mining, we agree that expanding this to other more complex domains like Weather, Epidemiology would further enrich its scope. Upon acceptance, we will make the full benchmark dataset publicly available and extend it with additional domains as future work to further strengthen its coverage and utility for the research community.

---

> ### Author Response · Authors · 2025-11-17
> **Addressing Weakness : W3 , Response to Question : Q1**
>
> ### **W3: Rationale for Baseline Selection**
> We thank the reviewer for this comment. STReason is positioned to address a capability gap that existing spatiotemporal models do not occupy, and we clarify our baseline choice accordingly.
>
> **Why existing models are not suitable baselines?** As discussed in Sections 1 and 2, prior spatiotemporal models are designed for numerical tasks (typically forecasting), evaluated with RMSE/MAE, and cannot process natural language, generate explanations, or perform multi-step reasoning. They operate on fixed tensors, not analytical queries. Thus, they cannot be meaningfully compared in a long-form QA setting.
>
> We explored three categories:
> - **Program-based Models (e.g., VISPROG, ViperGPT)**: These systems differ fundamentally from our setting as they operate over visual or symbolic inputs, produce object labels or binary decisions.  They lack numeric analytics and cannot execute STReason-style analytical pipelines (Appendix A.1).
> - **Domain-specific Spatiotemporal Models (e.g., STGCN, GraphWaveNet, AirFormer, AirPhyNet)**: These models are optimized for numeric prediction in specific domains and do not generate interpretable, long-form answers. They can serve as backend tools (as we do with GraphWaveNet) but not as baselines for language-driven reasoning.
> - **Spatiotemporal aware LLMs (e.g., UrbanGPT, UrbanLLM, UniST, ST-LLM+)**: Although these models incorporate LLMs, they remain focused primarily on numeric forecasting and do not support structured explanation or reasoning. The closest prior work, UrbanLLM, addresses urban QA but generates short, surface-level responses without  coherent long-form reasoning. Hence these models are also not directly comparable.
>
> **Why general-purpose LLMs are the right comparison group?** Due to the absence of existing systems capable of multi-task, long-form spatiotemporal reasoning, we compared STReason against strong general-purpose LLMs, which represent the strongest and scalable reasoning models currently available. All LLM baselines accept the same natural language input as STReason and attempt the same reasoning tasks. Comparing against LLMs isolates the contribution of our architecture, rather than comparing reasoning with models that were never designed to reason.
>
> In summary, we view **STReason as a foundational system for this emerging direction of spatiotemporal reasoning**, analogous to how models like  VISPROG and PAL advanced visual and mathematical reasoning. By extending the program-based reasoning paradigm to spatiotemporal analytics, we aim to establish a foundation for future tool-augmented or agent-style models.
>
> ### **Q1: How are Command Interpreter Modules determined?**
> We thank the reviewer for this question. The Command Interpreter Modules are not arbitrarily chosen; they follow a principled, task-driven design grounded in established spatiotemporal analytics and common analytical workflows.
>
> STReason defines 12 specialized modules, each corresponding to a analytical sub-task spanning the three core spatiotemporal task families (Section 3.2). Each module wraps a well-established statistical method or a SOTA pretrained model, ensuring reliability without requiring task-specific training for each query. For example:
> - *forecast()* module embeds a pretrained GraphWaveNet model
> - *analyze_trend()* module applies seasonal decomposition and linear regression
> - *detect_anomaly()* leverages unsupervised z-score based techniques
> - *gen_explanation()* uses an LLM guided approach to synthesize intermediate results into natural language.
>
> All module definitions are documented in Appendix A.4, with implementations openly available in step_funcs.py in our shared repository. The existing modules satisfy all three criteria below:
> - Widely used in real-world spatiotemporal analytics
> - Can be composed into multi-step reasoning pipelines
> - Represent reusable operations across multiple domains
>
> Importantly, these modules are interchangeable analytic backends, not fixed components. Modules can be added, replaced, or removed with only a few lines of declarative registration. For instance, GraphWaveNet in forecast() can be replaced by LSTM, DCRNN, or ST-LLM+ without altering the program generation process or modifying prompts. This demonstrates that the module set is minimal, compositional, and extensible.
>
> In summary, our design intentionally prioritizes reuse and integration of high-performing pretrained models, rather than retraining or fine-tuning for every new task, thereby reducing computational cost and increasing practicality. Hence, we believe **STReason demonstrates a principled and extensible approach to integrating LLMs with structured analytical pipelines/existing SOTA models** and opens new directions for scalable and interpretable, spatio-temporal AI.

---

> ### Author Response · Authors · 2025-11-24
>
> Dear Reviewer 2E98,
>
> We sincerely appreciate the time and effort you have dedicated to reviewing our work. We have carefully addressed your questions and concerns above and are happy to continue the discussion should you have any further questions or require clarifications.
>
> If there are no further questions, we kindly request you to consider revisiting your score in light of our responses, as we are approaching the end of the rebuttal period.
>
> Thank you once again for your valuable feedback and for helping us improve our work !

---

> ### Author Response · Authors · 2025-11-27
>
> Dear Reviewer 2E98,
>
> We would like to kindly follow up regarding our previous response to your comments.
>
> As the discussion period concludes on December 2, we would be grateful if you could consider revisiting your overall assessment  including both the score and the confidence rating in light of the responses we have provided. We sincerely appreciate your time and constructive feedback, and we remain happy to clarify any remaining concerns.
>
> Please let us know if you have any further questions or if additional clarification would be helpful from our side.
>
> Thank you once again for your valuable feedback and for helping us improve our work !

---

> > ### Comment · Reviewer_2E98 · 2025-11-27
> >
> > Thank you for the reminder. I'm satisfied with the response on W3 and Q1. However, the concern on generalizability (W1) is still not resolved, especially when all evaluations are on a very small dataset (W2). If the proposed approach is showing improvement on other publicly available benchmarks?

---

> ### Author Response · Authors · 2025-11-28
> **Clarification and Additional Empirical Evidence on STReason’s Generalizability**
>
> We thank the reviewer for the follow-up question and provide additional clarification and new experimental evidence to address concerns regarding generalizability.
>
> ### **Why dataset scale does not limit STReason’s generalization**
>
> As stated under W2 of our initial response, **no standardized benchmark currently exists for evaluating multi-task, long-form, interpretable spatiotemporal reasoning,** although there are existing benchmarks built for visual reasoning, logical/mathematical problem solving etc. (Appendix A.1). Existing spatiotemporal datasets (e.g., METR-LA, PEMS-BAY, BeijingAir, KnowAir) are generally used for numerical prediction tasks and do not contain natural-language queries or explanation ground truth. Hence, we constructed this benchmark dataset from scratch as the first systematic evaluation suite tailored for such reasoning tasks.
>
> Because STReason is training-free with no fine-tuning, no supervision, and no domain-specific adaptation, its capability does not depend on dataset size, unlike typical data-driven models. Instead, what matters is whether the reasoning pipeline transfers across:
> - different task families,
> - heterogeneous subtasks (trend, seasonality, neighborhood influence, anomalies, constraint enforcement, etc.), and
> - different real-world domains.
>
> As elaborated under W2, **our benchmark, although modest in size, is densely constructed** for this purpose. All 150 instances incorporate richly structured reasoning elements including temporal patterns, spatial dependencies, short/long-term forecasting, constraint enforcement and multi-step explanation across distinct domains (traffic and air quality). As reported in our main results, STReason consistently performs well across all these settings, demonstrating robustness to varied analytical structures across different domains.
>
> Crucially, STReason’s generalization does not depend on seeing large numbers of examples, but on its compositional program synthesis mechanism. This is further supported by the domain generalizability experiments in Appendix A.11.
>
> ### **Generalization Across Domains (Appendix A.11 + New Evaluation Datasets)**
>
> To directly address the reviewer’s concern regarding generalizability, we demonstrate STReason’s zero-shot capability through two complementary approaches:
> - Example Query based zero-shot generalization (Appendix A.11)
> - Dataset based zero-shot generalization (New Evaluation Datasets)
>
> Together, these provide a clearer  picture of STReason’s generalization ability.
>
> **1. Example Query based zero-shot generalization:** As reported in Appendix A.11, we evaluated STReason on individual queries from two domains completely unseen during system development, epidemiology and finance using public datasets (ILI Case Counts and Exchange Rate Data, Wu et al., 2022). Without modifying the architecture, prompts, or in-context examples, STReason:
> - generated correct and executable programs,
> - adapted arguments to domain-specific semantics,
> - produced coherent, domain-appropriate long-form explanations, and
> - satisfied all constraints and analytical requirements.
>
> This demonstrates true zero-shot generalization at the example level, showing that STReason’s reasoning pipeline is not tied to any domain-specific patterns.
>
> **2. Dataset based zero-shot generalization:** While Appendix A.11 illustrates generalizability through representative examples, the reviewer specifically asked for broader evaluation analogous to “comparison against other benchmarks.”
>
> Because no spatiotemporal reasoning benchmarks exist, **we constructed two new evaluation sets**, each containing 20 natural-language spatiotemporal reasoning queries, one for epidemiology and one for finance. These datasets were created independently of the original benchmark and were never used during system development.
>
> We then evaluated STReason against the strongest baseline from our original experiments (DeepSeek-V3). Consistent with our main results, **STReason achieved substantially higher Factuality Score, across both domains**:
>
> | | **Epidemiology Data** | **Finance Data** |
> | ------------------ | ----------------------- | -------------------------- |
> |   DeepSeek-V3     |          26.28%      |   43.33%                     |
> | STReason            | **83.05%**           | **87.5%**                 |
>
> These results provide dataset-level evidence that STReason generalizes robustly across domains, even when evaluated over an entire set of queries, not just isolated examples.

---

> ### Author Response · Authors · 2025-11-28
> **Clarification and Additional Empirical Evidence on STReason’s Generalizability (continued)**
>
> ### **Generalization to new  Task Types**
>
> STReason also generalizes to new analytical operations beyond the original tasks. To further illustrate STReason’s ability to handle entirely new analytical tasks, we included a new task type; correlation analysis, without modifying the model architecture or retraining. This required only registering a new function COMPUTE_CORRELATION(), demonstrating plug-and-play extensibility.
>
> Once added, the Command Generator immediately invoked the new function correctly and produced a full, long-form explanation. Below is an example query and the corresponding output produced by STReason:
>
> **Query:** *Perform an analysis of correlation between air quality and humidity over the last 30 days at the dongsi_aq in the Beijing area, focusing everyday. The current date and time are 2018-01-04 07:00:00, with data being recorded every 60 minutes. Please offer a detailed explanation.*
>
> **Execution Summary:** Loaded data for Location: dongsi_aq, Feature: air quality, Time Range: From 2017-12-05 07:00:00 to 2018-01-04 07:00:00. No constraints imposed Loaded weather data as temporal auxiliary data for Location: dongsi_aq, Time Range: From 2017-12-05 07:00:00 to 2018-01-04 07:00:00. Computed correlation between 'air quality' and 'humidity': 0.7967 Final Explanation Generated.
>
> **Final Answer:** Comprehensive Analysis of the Original Data: The original data consists of a time series of air quality measurements recorded at every 1 hour over a period of 30 days. The data ranges from a minimum value of 10.0 micrograms per cubic metre to a maximum value of 26.0 micrograms per cubic metre. The average air quality measurement across all hours is approximately 17.5 micrograms per cubic metre. The data appears to exhibit some variability, with fluctuations in air quality observed throughout the day.
>
> Analysis of Correlation with Humidity: The statistical correlation between air quality and humidity is calculated to be 0.7967. This indicates a strong positive correlation between air quality and humidity, suggesting that as humidity levels increase, air quality tends to deteriorate. This finding is important as it highlights the potential influence of humidity on air quality levels.
>
> Interpretation in Relation to Data Collection Constraints: Given that the data corresponds to everyday measurements, the correlation analysis between air quality and humidity can provide valuable insights into the relationship between these two variables on a daily basis. Understanding this correlation can help in identifying potential factors that contribute to changes in air quality levels and provide insights for air quality management strategies. Additionally, by analyzing the data over a 30-day period, it is possible to observe trends and patterns in air quality fluctuations, which may be influenced by factors such as weather conditions, traffic patterns, or industrial activities.
>
> ### **Summary of Response**
> - The benchmark’s scale does not constrain generalization.
> - The benchmark is dense, covering diverse reasoning factors across two domains.
> - Appendix A.11 already demonstrates strong example-level zero-shot transfer to new domains.
> - Two newly constructed datasets further provide dataset-level empirical evidence of robust generalization.
> - A new task type (correlation analysis) was seamlessly incorporated, showing task-level generalizability.
>
> Together, these results clearly demonstrate that **STReason generalizes well beyond its original evaluation suite, both across domains and analytical task types**, highlighting the robustness of its compositional, program-guided reasoning architecture.
>
> We hope these new results and clarifications address your remaining concerns and provide further evidence of STReason’s robustness and practical utility in real-world spatiotemporal systems.

---

### Official Review · Reviewer_pm73 · 2025-11-01

**Soundness:** 3
**Presentation:** 3
**Contribution:** 2
**Rating:** 2
**Confidence:** 3

**Summary:**

Traditional spatiotemporal models are specialized for narrow tasks (such as forecasting) and cannot handle complex reasoning, whereas general LLMs struggle to analyze ST data directly, often losing critical info when the data is converted to text. This paper presents a framework (STReason) to make LLMs into generalizable spatio-temporal reasoning systems. It uses in-context learning to decompose natural language queries into executable programs and executes these programs. A new benchmark dataset of 150 instances across three tasks (Analysis, Anomaly Detection, and Prediction & Reasoning) and an evaluation metric for long-form spatio-temporal reasoning are also proposed in the work.

**Strengths:**

1. Long-form reasoning for spatio-temporal data is an important and underexplored problem.
2. ST Reason's training-free approach makes it practically advantageous.

**Weaknesses:**

1. Only 150 examples across three tasks. This small scale makes it difficult to believe the STBench's robustness and generalizability. And Human evaluation is conducted only on 18 questions.
2. Only 12 pre-defined modules in the Function Pool limit generalizability, even though the paper claims "task and domain-agnostic". Function Pool requires manual intervention to add new modules.
3. Task-specific in-context examples require manual curation, which also limits generalizability.
4. The framework is benchmarked only against vanilla LLMs like (GPT-4, Deepseek-V3), they don't compare STReason with any other Spatio-temporalness aware LLMs. The experiment, as designed, demonstrates that "LLMs with ST tools" are more effective at ST tasks than "LLMs without ST tools," which is an expected outcome.

**Questions:**

1. How does it perform on completely new task types not seen in the three categories?
2. What is the performance if LLMs are given the ST models as tools? (No function pool or in-context examples)

---

> ### Author Response · Authors · 2025-11-17
> **Addressing Weaknesses : W1**
>
> ### **Summary of Response**
> We appreciate reviewer's recognition of the novelty of Long-form reasoning and the practical significance of our training-free apporach. We also thank the reviewer for the constructive feedback. In this response we clarify our rationale regarding dataset scale, function pool extensibility, the role of in-context examples, and baseline selection.
>
> ### **W1: Dataset Scale and Human Evaluation Coverage**
> We appreciate reviewer’s concern regarding dataset scale and robustness. As discussed in the manuscript, no standardized benchmark currently exists for evaluating multi-task, long-form, interpretable spatiotemporal reasoning. Unlike prior datasets, which typically focus on single-task numerical outputs, we constructed this benchmark dataset from scratch as the first systematic evaluation suite tailored for such reasoning tasks.
>
> **Benchmark Dataset:** Our benchmark was intentionally designed for methodological soundness rather than data volume, guided by the following principles:
> 1. **Designed for Breadth over quantity:** STReason is a training-free framework and does not learn parameters from data but evaluates how well a program-guided reasoning system generalizes across diverse spatiotemporal settings. Consequently, the benchmark’s purpose is not statistical representativeness but conceptual coverage, testing reasoning accuracy, constraint adherence, factuality, and interpretability across varied spatiotemporal conditions rather than optimizing performance on large sample sizes.
> 2. **Richness and diversity of each instance:** Each of the 150 instances couples (i) a natural-language query, (ii) a step-by-step executable program, and (iii) a verified ground truth  output. Constructing and validating these triples required extensive human annotation and cross-checking of intermediate reasoning steps, ensuring correctness and logical coherence at every stage. Each instance encodes multiple reasoning factors including spatial and temporal dependencies, forecasting horizons, and domain constraints making the dataset dense in reasoning diversity despite modest scale.
> 3. **Representative cross-domain coverage:** As described in Section 4.1, the dataset already spans three fundamental spatiotemporal tasks across two real-world domains: land traffic (PEMS-BAY and METR-LA) and Air quality (Beijing and Shenzhen). These domains were chosen because they embody distinct temporal dynamics and spatial dependencies, forming a robust foundation for evaluating cross-domain generalization. Moreover, Appendix A.11 demonstrates the generalizability of STReason to other unseen domains, further reinforcing that the benchmark already captures essential diversity in data and reasoning structure.
>
> While our dataset covers two major domains and three of the most representative tasks in Spatiotemporal Data Mining, we agree that expanding to other more complex domains like Weather, Epidemiology would further enrich its scope. We will make the full benchmark dataset publicly available upon acceptance. We also plan to extend it with additional domains as future work to further strengthen its utility for the research community.
>
> **Human Evaluation:** The human study was intentionally limited to 18 representative queries as **Long-form reasoning is cognitively intensive.** Each query required evaluators to assess multi-step analytical workflows, constraint enforcement, spatial and temporal attribution, and numerical reasoning, NOT simple short answers or single-sentence factual checks.
>
> Evaluators compared responses generated by STReason with those from baseline LLMs, focusing on multi-step reasoning quality, factual grounding, coherence, and interpretability. To achieve reliable judgments, 27 independent evaluators collectively conducted 486 pairwise comparisons (27 evaluators × 18 queries). **This scale was chosen to balance statistical reliability with cognitive feasibility, given the depth of reasoning each answer required.**
>
> Automatic metrics including Constraint Adherence, Factuality, and Coherence were computed for all 150 instances in the benchmark, while the human study served as an additional validation layer. As mentioned in Section 4.3, **the results showed a strong alignment between automatic and human evaluations**, confirming that the proposed metrics effectively capture human-perceived reasoning quality.

---

> ### Author Response · Authors · 2025-11-17
> **Addressing Weaknesses : W2**
>
> ### **W2: Function Pool Generalizability and Task/Domain Agnosticism**
>
> We thank the reviewer for this valuable comment on the Function Pool design and its connection to task and domain generalizability. Please find our clarifications below:
>
> **1. Why STReason is Domain and Task Agnostic?**
>
> **Domain agnostic Design:** Unlike existing spatiotemporal models (UrbanLLM, UrbanGPT, ST-LLM+, UniST etc.), which are confined to a single domain (typically traffic) and require retraining per domain dataset, STReason operates across different domains using one unified framework. The framework has already been evaluated on traffic and air-quality related queries and as demonstrated in Appendix A.11, it generalizes zero-shot to entirely new domains (finance and epidemiology) without any architectural or prompting changes. This confirms that STReason’s reasoning and execution mechanisms are domain-independent, relying solely on structured input data rather than domain-specific fine-tuning. We plan to extend this ability to multimodal spatiotemporal domains as part of future work (Section 5).
>
> **Task Agnostic Architecture** STReason supports multiple analytical task families; Analysis, Anomaly Detection, and Prediction & Reasoning, within a single architecture. By contrast, traditional deep-learning frameworks (e.g., STGCN, ASTGCN, GMAN, STID) and most LLM-based baselines (ST-LLM+, UniST) are typically trained for a single predictive task and cannot handle heterogeneous operations. To support new tasks beyond those defined in our Function Pool, users only need to define a small function and register it using the standardized module interface, a process that typically takes only a few minutes. In contrast, existing spatiotemporal baselines require significant engineering effort, domain expertise, substantial training data, and hours/days of fine-tuning to support new tasks. STReason avoids this overhead entirely, offering a lightweight and practical mechanism for adding new analytical capabilities.
>
> In summary, STReason’s modular architecture enables scalable, low-effort adaptation to new tasks, domains, and models, rather than fine-tuning separate models from scratch which is often computationally expensive and time-consuming. This plug-and-play flexibility minimizes engineering effort while enabling rapid deployment within a unified, interpretable reasoning framework.
>
> **2. Function Pool design and extensibility**
>
> **Existing Function Pool Design:** STReason’s Function Pool is designed as a declarative and modular registry that maps each function to its natural language description, argument schema, and output format. We agree that the current Function Pool includes only 12 pre-defined modules. However, this compactness is a design choice rather than a limitation.
>
> Rather than maintaining a large, static repository of rarely used functions, STReason uses a plug-and-play mechanism where new analytical functions can be added with just a few lines of declarative configuration (Appendix A.2). Once registered, the Command Generator immediately recognizes and invokes new modules through in-context reasoning. This design ensures the **Function Pool is lightweight, compositional, and easily extensible.**
>
> **Function Pool Extensibility:** We acknowledge that as STReason continues to expand to support additional tasks, manual maintenance of the Function Pool may become a scalability bottleneck. We have therefore included automated Function Pool maintenance as part of our future work (Section 5) to further enhance scalability and reduce manual intervention. We specifically plan to explore the following strategies:
> - **LLM-Guided Function Induction:** Use the LLM to automatically inspect interpreter libraries and infer new function definitions, allowing dynamic Function Pool expansion without human intervention.
> - **Usage-Based Pruning:** Track invocation statistics to identify and remove under-utilized functions, keeping the pool efficient and adaptive.
>
> These are high-level directions that would require further research and validation, but we believe they highlight promising avenues for improving long-term scalability and usability.
>
> Meanwhile, we also note that the current declarative format of the Function Pool supports simple, scalable updates where new functions can be added or modified with minimal engineering effort, and without changes to the core program generation or execution logic. We believe this design is a strong foundation for incorporating automated maintenance tools in the future.

---

> ### Author Response · Authors · 2025-11-17
> **Addressing Weaknesses : W3**
>
> ### **W3: Curation of In-Context Examples and Generalizability**
>
> We thank the reviewer for raising this concern. As noted in Section 5, STReason currently relies on manually curated in-context query–program pairs. We acknowledge that this process could limit scalability to entirely new task types; however, **it does not constrain generalization to new domains.**
>
> **Domain Generalization:** To verify domain-agnosticity, we tested STReason on unseen domains without modifying either the architecture or the existing in-context examples. The model successfully generated structured and executable programs that adapted arguments and semantics to each new setting:
>
> **Example 1 : Financial Domain**
> **Query:** Analyze the trend in the historical stock price data for the past 12 hours for ticker symbol AAPL on the NASDAQ exchange, focusing on weekdays. The current time is 2023-10-12 15:00:00. The data is recorded at 5-minute intervals. Analyze the trend significance.
>
> **Generated Program:**
> ```
> DATA0 = LOAD_SPATIOTEMPORAL_DATA(ticker="AAPL", exchange="NASDAQ", time="2023-10-12 15:00:00", feature='stock price', time_int=5, period=12, unit="hours", task="analysis")
> DATA0_CONST = IMPOSE_CONSTRAINTS(data=DATA0, time="2023-10-12 15:00:00", time_int=5, period=12, unit="hours", task="analysis", constraint="weekdays only", constraint_val="None", preds="None")
> TREND_ANALYSIS = ANALYZE_TREND(data=DATA0_CONST, time_int=5, feature='stock price', ticker="AAPL", constraint="weekdays only")
> EXPLANATION = GEN_EXPLANATION(task="analysis", data=DATA0_CONST, feature='stock price', ticker="AAPL", exchange="NASDAQ", time_int=5, horizon="None", horizon_unit="None", constraint="weekdays only", constraint_val="None", trend=TREND_ANALYSIS, seasonality="None", neighbourhood="None", preds="None", sensitivity="None")
> FINAL_RESULT = REFINE_OUTPUT(var=EXPLANATION)
> ```
>
> **Example 2 : Epidemiology Domain**
> **Query:** The current time is 2022-11-01 08:00:00. Predict the flu incidence rate at hospital ID 872 in the Seattle region for the next 48 hours using historical data from the past 7 days, recorded at hourly intervals. Ensure the predicted incidence rate does not exceed the 15% outbreak threshold. Analyze how time of day and neighboring hospitals influence the prediction.
>
> **Generated Program:**
> ```
> DATA0 = LOAD_SPATIOTEMPORAL_DATA(location="872", time="2022-11-01 08:00:00", feature='flu incidence rate', region="Seattle", time_int=60, period=7, unit="days", task="prediction")
> PREDICTION = FORECAST(data=DATA0, location="872", time="2022-11-01 08:00:00", feature='flu incidence rate', region="Seattle", time_int=60, period=7, unit="days", horizon=48, horizon_unit="hours")
> PREDICTION_CONST = IMPOSE_CONSTRAINTS(data=DATA0, time="2022-11-01 08:00:00", time_int=60, period=7, unit="days", task="prediction", constraint="outbreak threshold", constraint_val=15, preds=PREDICTION)
> SPATIOTEMPORAL_SENSITIVITY = CONDUCT_SENSITIVITY_ANALYSIS(data=DATA0, preds=PREDICTION_CONST, location="872", time="2022-11-01 08:00:00", feature='flu incidence rate', region="Seattle", time_int=60, period=7, unit="days", horizon=48, horizon_unit="hours")
> EXPLANATION = GEN_EXPLANATION(task="prediction", data=DATA0, feature='flu incidence rate', location="872", region="Seattle", time_int=60, horizon=48, horizon_unit="hours", constraint="outbreak threshold", constraint_val=15, trend="None", seasonality="None", neighbourhood="None", preds=PREDICTION_CONST, sensitivity=SPATIOTEMPORAL_SENSITIVITY)
> FINAL_RESULT = REFINE_OUTPUT(var=EXPLANATION)
> ```
> Program generation for similar queries can be tested through the STtask_exec.ipynb notebook provided in our code repository.
>
> **Role of In-context Examples:** We would also like to emphasize that STReason’s reasoning ability arises from its structured architecture, not from memorization of task-specific examples. The Function Pool and Command Interpreter drive program synthesis by grounding each operation in explicit functional definitions while the in-context examples serve only as task-level templates that demonstrate how to compose functions, rather than as repositories of domain or dataset knowledge.
>
> Empirically, our ablation study (Table 3) confirms this distinction. We conducted a Test-Query-Exclude ablation, where all in-context examples sharing the same task type as the test query were deliberately removed. Even under this strict setting, STReason continued to produce correct and executable programs, maintaining high program generation accuracy. This shows that the framework can infer the correct reasoning flow purely from functional semantics, without relying on task-aligned examples.
>
> In summary, **in-context examples provide format and composition cues, while the structured Function Pool encodes the key reasoning semantics** ensuring stable performance even with minimal or generic examples. This also highlights STReaon's scalability and robustness beyond manually curated prompts.

---

> ### Author Response · Authors · 2025-11-17
> **Addressing Weaknesses : W4**
>
> ### **W4: Baseline Scope and Comparison with Spatiotemporal-Aware LLMs**
> We appreciate the reviewer’s comment regarding the choice of baselines. Below we clarify our rationale and position STReason within the broader landscape of spatiotemporal reasoning models.
>
> **Motivation for STReason:** As elaborated in Section 1, the primary motivation for developing STReason is the lack of spatiotemporal models capable of interpretable, long-form question answering and reasoning. Existing studies cited in Section 2 are largely designed for narrow, task-specific objectives and face two core limitations:
> - **Limited to numeric forecasting tasks:** Most prior models are evaluated only on scalar outputs (e.g., RMSE or MAE for prediction tasks). They are not designed to produce or assess multi-step reasoning quality or explanatory outputs (Lines 46–50)
> - **Lack of language-driven reasoning:** These frameworks operate on fixed-format spatiotemporal tensors and cannot process natural language queries. Consequently, their evaluation assumes structured numeric inputs/outputs and omits interpretability or linguistic reasoning.
> (Lines 53–57)
>
> These limitations highlight the absence of existing systems capable of multi-task, language-driven, spatiotemporal reasoning, which is precisely the gap STReason aims to fill.
>
> **Related Models Explored for Comparison:** Accordingly, after developing STReason, we examined existing approaches across three categories to ensure a comprehensive comparison:
> - **Program-based Models (e.g., VISPROG, ViperGPT)**: These systems differ fundamentally from our setting as they operate over visual or symbolic inputs, produce object labels or binary decisions, and lack numerical analytics or long-form explanation capabilities. As detailed in Appendix A.1, direct comparison is not feasible due to mismatched modalities, query types, and output formats.
> - **Domain-specific Spatiotemporal Models (e.g., STGCN, GraphWaveNet, AirFormer, AirPhyNet)**: These models are optimized for numeric prediction in specific domains and do not generate interpretable, long-form answers. While such models can serve as analytical backends within STReason's interpreter library, they cannot serve as meaningful baselines for evaluating language-based, multi-task reasoning.
> - **Spatiotemporal aware LLMs (e.g., UrbanGPT, UrbanLLM, UniST, ST-LLM+)**: Although these models incorporate LLMs, they remain focused primarily on numeric forecasting and do not support structured explanation or reasoning. The closest prior work, UrbanLLM, addresses urban QA but generates short, surface-level responses without  coherent long-form reasoning. Hence these models are also not directly comparable.
>
> **Choice of Baselines:** Accordingly, due to the absence of existing systems capable of multi-task, long-form spatiotemporal reasoning, we compared STReason against strong general-purpose LLMs, which represent the most capable and scalable reasoning models currently available. Across all baselines, STReason delivers consistent gains in factuality, coherence, and constraint adherence, demonstrating that these improvements arise from our architectural design, not merely from “LLMs + tools.”
>
> That said, we view **STReason as a foundational system for this emerging direction of spatiotemporal reasoning**, analogous to how models like  VISPROG and PAL advanced visual and mathematical reasoning. By extending the program-based reasoning paradigm to spatiotemporal analytics, **we aim to establish a foundation for future tool-augmented or agent-style spatiotemporal models.** We will publicly release our benchmark and evaluation code to facilitate direct comparison with forthcoming models that integrate structured reasoning and domain-specific tools.

---

> ### Author Response · Authors · 2025-11-17
> **Response to Questions : Q1**
>
> ### **Q1: How does STReason perform on completely new task types not seen in the three categories?**
> We thank the reviewer for this question. We clarify that STReason already supports a broad spectrum of analytical subtasks and is explicitly designed to extend to new task types with minimal effort.
>
> **Broad coverage of task types:** Although we group tasks into three families (Analysis, Anomaly Detection, Prediction & Reasoning), each family includes multiple fine-grained operations including trend/seasonality analysis, neighbourhood influence, anomaly detection and impact, short/long-term forecasting, sensitivity analysis, constraint enforcement, and multi-step explanation. These subtasks arise naturally in real-world spatiotemporal QA, and our experiments show that STReason can flexibly combine them into coherent reasoning pipelines.
>
> **Zero-shot generalization to unseen tasks in new domains:** As shown in Appendix A.11, STReason generalizes to unseen domains without modifying the architecture, prompts, or examples. The model produced correct, executable programs for exchange-rate prediction and ILI-case analysis, capturing domain semantics and constraints despite never encountering these task types.
>
> **Adding New task types with minimal effort:** For task types not currently included, STReason supports rapid extensibility due to its plug-and-play Function Pool design elaborated under W2:
> - Adding a new operation requires only registering a new function in the Function Pool with a simple input/output schema.
> - Once registered, the Program Generator immediately begins to invoke the new function correctly, often with only 1–2 in-context examples
>
> No retraining, fine-tuning, or architectural changes are needed. Hence our design contrasts with both traditional and LLM-based spatiotemporal models, which typically require new architectures, domain-specific retraining or substantial feature engineering to support new tasks and still cannot produce long-form reasoning or interpretable explanations.
>
> **Example Demonstration of a New Task Type: Correlation Analysis** To further illustrate STReason’s ability to handle entirely new analytical tasks, we included a new task type: correlation analysis, without modifying the model architecture or retraining. This required only registering a new function *COMPUTE_CORRELATION()*, demonstrating plug-and-play extensibility.
>
> Below is an example query and the corresponding output produced by STReason:
>
> ***Query:** Perform an analysis of correlation between air quality and humidity over the last 30 days at the dongsi_aq in the Beijing area, focusing everyday. The current date and time are 2018-01-04 07:00:00, with data being recorded every 60 minutes. Please offer a detailed explanation.*
>
> **Execution Summary:**
> Loaded data for Location: dongsi_aq, Feature: air quality, Time Range: From 2017-12-05 07:00:00 to 2018-01-04 07:00:00.
> No constraints imposed
> Loaded weather data as temporal auxiliary data for Location: dongsi_aq, Time Range: From 2017-12-05 07:00:00 to 2018-01-04 07:00:00.
> Computed correlation between 'air quality' and 'humidity': 0.7967
> Final Explanation Generated.
>
> **Final Answer**:
> Comprehensive Analysis of the Original Data: The original data consists of a time series of air quality measurements recorded at every 1 hour over a period of 30 days. The data ranges from a minimum value of 10.0 micrograms per cubic metre to a maximum value of 26.0 micrograms per cubic metre. The average air quality measurement across all hours is approximately 17.5 micrograms per cubic metre. The data appears to exhibit some variability, with fluctuations in air quality observed throughout the day.
>
> Analysis of Correlation with Humidity: The statistical correlation between air quality and humidity is calculated to be 0.7967. This indicates a strong positive correlation between air quality and humidity, suggesting that as humidity levels increase, air quality tends to deteriorate. This finding is important as it highlights the potential influence of humidity on air quality levels.
>
> Interpretation in Relation to Data Collection Constraints: Given that the data corresponds to everyday measurements, the correlation analysis between air quality and humidity can provide valuable insights into the relationship between these two variables on a daily basis. Understanding this correlation can help in identifying potential factors that contribute to changes in air quality levels and provide insights for air quality management strategies. Additionally, by analyzing the data over a 30-day period, it is possible to observe trends and patterns in air quality fluctuations, which may be influenced by factors such as weather conditions, traffic patterns, or industrial activities.

---

> ### Author Response · Authors · 2025-11-17
> **Response to Questions : Q2**
>
> ### **Q2: What is the performance if LLMs are given the ST models as tools? (No function pool or in-context examples)**
>
> We appreciate the reviewer’s question. In short, simply giving an LLM access to spatiotemporal (ST) models does not produce reliable results. This is precisely why STReason is built around structured function definitions, argument schemas, and program synthesis, rather than exposing raw tools. We clarify this below:
>
> **LLMs cannot reliably use ST models without structured interfaces** LLMs can read, understand and describe code, but they cannot execute code or reliably orchestrate complex spatiotemporal models without a structured interface. Specifically advanced spatiotemporal models such as GraphWaveNet, DCRNN, etc., require highly parameterized inputs (sequence length, prediction horizon, adjacency matrix etc.). If these models are exposed “as-is” to an LLM:
> - the prompt must include all model code + all data, which potentially exceeds context limits
> - LLMs hallucinate function signatures or parameter names
> - LLMs describe steps instead of producing executable outputs
> - the LLM cannot bind arguments correctly
>
> These failure cases are similar to what we observed in our experiments as shown in Appendix A.10. Even advanced LLMs often produced incorrect temporal and spatial reasoning, hallucinated model components and produce descriptions without actual execution. Thus, “LLM + raw ST models” is not a viable reasoning pipeline.
>
> **Why STReason’s program-guided pipeline is necessary**
> In contrast to giving LLMs raw access to spatiotemporal models, STReason is intentionally designed around a structured, program-guided reasoning pipeline composed of four tightly integrated components:
> - **In-Context Examples:** Provide format and compositional cues that show how to structure multi-step analytical workflows.
> - **Function Pool:** Provides a controlled abstraction layer where each function has a fixed, compact signature, inputs and outputs follow a declarative schema, natural-language descriptions guide program synthesis, and every function maps to a valid analytical operation.
> - **Command Generator:** Synthesizes a coherent, step-wise program by binding arguments correctly, selecting the appropriate functions, and ordering analytical steps logically.
> - **Command Interpreter** Executes the generated program using backend analytical models.
>
> Together, this pipeline provides the LLM with a structured, unambiguous interface for orchestrating complex spatiotemporal analytics with correct argument binding, ordering of analytical operations, executable program synthesis andconsistent handling of spatiotemporal constraint.
>
> This combination makes STReason fundamentally different from simply “LLMs with tools”, enabling robust, multi-step, constraint-aware spatiotemporal reasoning that raw tool access cannot achieve.

---

> ### Author Response · Authors · 2025-11-24
>
> Dear Reviewer pm73,
>
> We sincerely appreciate the time and effort you have dedicated to reviewing our work. We have carefully addressed your questions and concerns above and are happy to continue the discussion should you have any further questions or require clarifications.
>
> If there are no further questions, we kindly request you to consider revisiting your score in light of our responses, as we are approaching the end of the rebuttal period.
>
> Thank you once again for your valuable feedback and for helping us improve our work !

---

> > ### Comment · Reviewer_pm73 · 2025-11-27
> > **Response to Authors**
> >
> > I thank the authors for their response. They have addressed my concerns, and I have updated my review score accordingly.

---

> > > ### Author Response · Authors · 2025-11-27
> > > **Note of Thanks**
> > >
> > > Dear Reviewer pm73,
> > >
> > > Thank you very much for your thoughtful engagement with our work and for taking the time to update your score. We are glad to hear that our responses have addressed your concerns.

---

### Author Response · Authors · 2025-12-01
**Meta-summary for AC**

We thank all reviewers for their constructive feedback. We are encouraged that the reviewers recognized the novelty, clarity, methodological significance, and strong empirical performance of STReason, as well as its interpretability, and robustness against strong LLM baselines. Below we summarize the main contributions of the paper, the key concerns raised, and how our rebuttal and new experiments address them.

### **Novelty & Contribution:**
STReason is a training-free, program-based framework that integrates the reasoning strengths of large language models (LLMs) with the analytical capabilities of spatiotemporal (ST) models for multi-task inference and execution. Our key contributions include:

- A unified architecture for long-form, multi-step ST reasoning.
- A new benchmark with natural-language queries, program traces, and long-form ground truth.
- Structured evaluation metrics for long-form ST reasoning.

Reviewers consistently noted our strong factuality gains, practical training-free design, transparent implementation, extensible architecture, and novel extension of program-based reasoning to ST tasks.

### **Summary of Concerns & Responses:**

**1.Dataset Scale & Generalization:** Reviewers (R1, R2, R4) raised concerns on generalizability of STReason due to the size of the benchmark. We clarified that STReason is training-free and therefore not constrained by dataset size. We further highlighted that each instance encodes multiple reasoning factors making the dataset dense in reasoning diversity despite modest scale. To empirically validate generalization, we demonstrated example-level zero-shot transfer to two unseen domains (Appendix A.11) and dataset-level evidence using two newly constructed evaluation sets in finance and epidemiology domains, where STReason substantially outperformed the strongest baseline (DeepSeek-V3) consistent with our main findings. These results indicate that generalization stems from STReason’s compositional architecture rather than from dataset size.

**2. Function Pool & In-context Examples:** Reviewers (R1, R3, R4) raised concerns on the role of in-context examples and the function pool. We clarified that in-context examples provide only format/composition cues, while the Function Pool is the true source of semantic grounding (argument schemas, descriptions, and valid operations). Function pool is lightweight and easily extensible, enabling reliable program synthesis. Ablations (Table 3) show that even when all task-type examples are removed, STReason still produces correct executable programs, demonstrating that its reasoning derives from functional semantics, not example memorization. Further, cross-domain experiments (Appendix A.11) reinforce this finding. We also outlined future directions for automating Function Pool expansion using LLM-guided function induction/usage-based pruning.

**3. Baseline Choice:** Reviewers questioned the absence of domain-specific and hybrid baselines. We clarified that:
- Existing ST models (e.g., STGCN, GraphWaveNet, AirFormer) are numeric predictors and cannot process natural-language queries or produce long-form reasoning.
- Hybrid LLM+tool systems (VISPROG, PAL) operate in incompatible modalities and cannot execute STReason-style analytical queries (Appendix A.1).
- ST-aware LLMs (UrbanGPT, UniST, ST-LLM+) focus on forecasting and lack multi-step analytical reasoning capabilities.

Since no existing system performs multi-task, natural-language ST reasoning, strong general-purpose LLMs (GPT-4, DeepSeek-V3) are the only meaningful baselines for our setting.

**4. Benchmark & Evaluation Protocol:** Reviewers (R3, R4) raised concerns on contribution of the benchmark and reliability of LLM-based evaluations. We clarified that no existing ST dataset contains natural-language queries, program traces, and reasoning ground truth. Our benchmark fills this gap and establishes the first standardized testbed for long-form ST reasoning. We also detailed our structured, deterministic evaluation pipeline and showed strong alignment between automated metrics and human judgments, supporting the validity of LLM-based evaluations.

**Summary:**

- Reviewer pm73 raised their score from 2 to 6 after the rebuttal.
- Reviewer 2E98 confirmed satisfaction on baselines and modules, and we additionally provided new cross-domain results to resolve  remaining concerns on generalization.
- Reviewer PRfp’s questions on baselines and manual components were fully addressed.
- Reviewer 69yp acknowledged our clarifications, and we provided a brief recap together with additional empirical evidence.

We are grateful for the reviewers’ insights. We believe the clarifications and additional experiments strengthen the contribution, demonstrating that STReason is a robust, extensible, and foundational framework for spatiotemporal reasoning research.

---

### Meta-Review · Area_Chair_zbkp · 2026-01-03

**Summary:**

This paper proposes a modular framework that combines LLMs with spatio-temporal models for multi-task reasoning. While the engineering effort is solid and the system is reasonably implemented, the overall contribution is incremental and closely follows an established LLM-as-planner with tool/function calling paradigm. The presentation is adequate but slightly below the typical standard, and the evaluation protocol is overly simple. Multiple reviewers raised concerns that the experimental evidence is insufficient to convincingly demonstrate the necessity and impact of the proposed design choices, particularly given the small benchmark size and limited baseline comparisons.

**Reviewer Concerns:**

There is strong agreement across reviewers on several key weaknesses: 1) limited novelty beyond prior modular or tool-augmented LLM frameworks, 2) an overly small and narrow evaluation benchmark, 3) overly simple or incomplete evaluation protocols, and 4) unclear empirical justification for the function pool and function-calling design.

While the rebuttal effectively addressed most concerns raised by reviewer pm73, it only partially addressed the concerns of the other reviewers. In particular, the core issues emphasized by reviewer 69yp (regarding whether the framework provides a substantive conceptual or empirical advance beyond existing approaches) remain insufficiently resolved in my view. Several concerns (e.g., evaluation strength, baseline coverage, and the actual benefit of the function pool) were shared across multiple reviews and persist even after rebuttal.

**Reviewer Scores:**

The final reviewer scores reflect a mixed but overall below-threshold consensus. Although reviewer pm73 increased their score after the rebuttal, the remaining reviewers maintained marginal or below-acceptance scores. It is unlikely that reviewers who remained critical would have significantly increased their scores had full discussion been possible. Taken together, the score distribution and discussion support a rejection decision.

---

### Decision · Program_Chairs · 2026-01-26

Reject